# Mineralogical and elemental geochemical characteristics of Taodonggou Group mudstone in Taibei Sag, Turpan-Hami Basin: Implication for its formation mechanism

Huan Miao[1,2*], Jianying Guo[3*], Yanbin Wang[4], Zhenxue Jiang[1,2], Chengju Zhang [1,2], Chuanming Li[1,5]

[1]State Key Laboratory of oil and gas resources and exploration, Beijing 102249, China;

[2]Institute of unconventional oil and gas science and technology, China University of Petroleum (Beijing), Beijing 102249, China;

[3] CNPC Key Laboratory of Natural Gas Accumulation and Development,Langfang 065007,China;

[4]School of Geosciences and Surveying Engineering,China University of Mining and Technology(Beijing),Beijing 100083,China;

[5]College of Geosciences, China University of Petroleum (Beijing), Beijing 102249, China;

*Correspondence to*: Huan Miao(1627765379@qq.com); Jianying Guo (gjy_17711224@petrochina.com.cn)

**Abstract.** Organic matter types in the Taodonggou Group mudstone exhibit significant differences with depth. In order to understand the formation mechanism of this special phenomenon, we analyzed the mineralogy and geochemistry of the mudstone, as well as the source rocks, depositional environment, and depositional processes of the Taodonggou Group. Based on this, we have gained the following understanding: (1) The Taodonggou Group mudstone was deposited in an intermediate-depth or deep, dyoxic, freshwater-brackish lake environment under warm and humid paleoclimatic conditions. The input of terrestrial debris was stable, but the sedimentation rate was slow. In addition, the sedimentation in the middle stage was influenced by hydrothermal activities, and the changes in the depositional environment corresponded to variations in organic matter types. (2) The source rocks of the Taodonggou Group mudstone are mainly andesitic and feldspathic volcanic rocks. Sediment sorting and recycling were weak, and hydrocarbon source information was well preserved. The tectonic background of the source area was a continental island arc and an oceanic island arc. Furthermore, changes in the provenance of the Taodonggou Group also had a significant impact on the variations in organic matter types. (3) The sedimentation of the Taodonggou Group involved both traction and gravity flows. The variations in source area, depositional environment, and depositional processes during different depositional periods led to changes in the organic matter types of the Taodonggou mudstone. (4) Based on the depositional environment, provenance, and depositional processes, the sedimentation of the Taodonggou Group can be divided into three stages. In the early stages, the sedimentation center was in the Bogda area. At this time, the Bogda Mountain region was not exposed, and the depositional processes inherited the characteristics of Early Permian gravity flow sedimentation, resulting in the widespread deposition of a series of high-quality Type III source rocks in the basin. In the middle stage of the Taodonggou Group sedimentation, the sedimentation center gradually migrated to the Taibei Sag. During this period, the Bogda Mountain region experienced uplift and hydrothermal activity, and the depositional processes gradually transitioned to traction flows, resulting in the widespread deposition of a series of Type II source rocks in the basin. In the late stage of the Taodonggou Group, the uplift of the Bogda Mountain region ceased, and the sedimentation

center completely shifted to the Taibei Sag. Meanwhile, under the influence of gravity flows, the organic matter types of the Taodonggou mudstone changed to Type III.

**Keyword:** Turpan-Hami Basin;Taodonggou Group;Mineralogy;Element Geochemistry;Sedimentary Environment

## 1 Introduction

Turpan-Hami Basin, Junggar Basin and Bogda area all belong to the southern part of the ancient Asian ocean in the Paleozoic era (Korobkin and Buslov, 2011; Jiang et al., 2015). During the Early Carboniferous to Early Permian, they began momentously to separate due to the continuous expansion of the Bogda Rift and began to enter the basin-forming period in the Middle Permian (Miao et al., 2004; Novikov, 2013; Jiang et al., 2015; Wang et al., 2019; Zhang et al., 2019). The Middle Permian is a momentous stage in the tectonic evolution of the Turpan-Hami basin. During this period, the expansion of the Bogda Rift stopped. With the gradual withdrawal of seawater from Xinjiang, the sedimentary environment of the Turpan-Hami basin gradually shifted to continental facies, and the sedimentary center gradually shifted from the Bogda area to Taibei Sag (Miao et al., 2004; Shi et al., 2020; Li et al., 2022). Taodonggou Group mudstones are widely deposited in the Turpan-Hami Basin. Previous studies have confirmed that Taodonggou Group mudstone is a very good to excellent source rock with huge hydrocarbon generation potential (Song et al., 2018; Miao et al., 2021; Miao et al., 2022; Miao et al., 2022a). It has been found that the organic matter types of the Taodonggou mudstone can be classified into two categories, with the upper and lower sections being Type III and the middle section being Type II (Miao et al., 2021; 2023).

The hydrocarbon generation potential of mudstone is closely related to its sedimentary environment (Wu et al., 2021; Li et al., 2022; Zhang et al., 2019; Zhao et al., 2021; Miao et al., 2004). Regarding the sedimentary environment of the Taodonggou Group mudstone, previous researchers have conducted extensive research. Miao et al. (2004) believed that the mudstone in the Taodonggou Group was deposited in a warm and humid paleoclimate, high-salinity water bodies, and an anoxic environment. Yang et al. (2010), based on the sedimentary characteristics of the Taerlang Formation and the Daheyan Formation, believed that the Taodonggou Group was deposited in a subhumid climate and that climate change is periodic. Wei (2015) also confirmed that the paleoclimate change of the Taodonggou Group stratum has a cyclical feature through tree rings and is mainly a warm and humid paleoclimate. At the same time, Song et al. (2018) also confirmed this by using the elemental geochemical characteristics of the Taodonggou Group shale outcrops in the field; Tian et al. (2017) analyzed the biomarkers of the Taodonggou Group in 7 outcrops around the Turpan-Hami Basin and concluded that the mudstone of the Taodonggou Group was deposited in a balanced, filled lake with little or no terrestrial organic matter, a large amount of algal organic matter input, and weakly alkaline, hypoxic to hypoxic brackish water. Miao et al. (2021) found biomarkers in the Taodonggou Formation mudstone from wells YT1 and L30 from different perspectives of Tian, which may be related to the weathering effect of outcrop samples. Through the research of the above scholars, we have found that there is some

controversy over the sedimentary environment of the Taodonggou Group, and the relationship between the cyclic changes in
the sedimentary environment and the changes in the organic matter types of the Taodonggou Group mudstone is still unclear.
In addition, the provenance and sedimentation mode of sediments also have a significant influence on the organic matter
types in mudstones (Mei et al., 2020). Mudstone belongs to a category of fine-grained sediment that is challenging to analyze
using traditional heavy mineral analysis methods (Rollinson, 1993; Roser and Korsch, 1988; Gehrels et al., 2008). Therefore,
elemental geochemical methods can be employed for provenance analysis (McLennan et al., 1983; Kröner et al., 1985; Li et
al., 2020). Elemental geochemical analysis compares the major, trace, and rare earth element characteristics of mudstones in
the sedimentary area with those of lithologies in the provenance area to determine the lithology of source rocks, weathering
degree, and tectonic background of the sediment source area (Li et al., 2020; Floyd and Leveridge, 1987; Basu et al., 2016).
Previous studies have found that the sediment source not only affects variations in the salinity of lake water but also influences
the input of nutrients and terrestrial organic matter, thus impacting the quality of mudstones (Li et al., 2020; Deditius, 2015;
Essefi, 2021). The tectonic activity in the source area not only affects changes in the sedimentary center but also influences
the source area (Miao et al., 2022b; Pinto et al., 2010). Therefore, reconstructing the location and sedimentation mode of the
sediment source area is of great significance for understanding the variations in organic matter types in the Taodonggou Group
mudstones.
Based on the mineralogical and elemental geochemical characteristics of 16 mudstone samples collected from the YT1
well, this study aims to reconstruct the paleoclimatic features, provenance, and tectonic background of the sedimentary period
in the source area of the Taodonggou Group mudstones. It also aims to explore the influence of sedimentary environment,
provenance changes, and sedimentation mode on the deposition of the Taodonggou Group mudstones, in order to reveal the
formation process of the mudstones.
**2 Geological setting**
The Turpan-Hami Basin, located in the eastern part of Xinjiang Uygur Autonomous Region, is one of the three major
petroliferous basins in Xinjiang. It is 660 km long from east to west and 130 km wide from north to south, with a total covered
area of $5.35 \times 104$ km$^2$. The Turpan-Hami Basin has undergone four stages: the extensional rift basin development stage, the
compressional foreland basin development stage, the extensional faulted basin development stage, and the compressional
regenerated foreland basin development stage, which finally formed the current pattern of the Mesozoic-Cenozoic
superimposed composite inland basin (Zhu et al., 2009; Jiang et al., 2015; Wartes et al., 2002; Greene et al., 2005). According
to the tectonic evolution characteristics of the Turpan-Hami Basin, the Turpan-Hami Basin can be divided into three primary
tectonic units from east to west: the Hami Depression, the Liaodun Uplift, and the Turpan Depression (Miao et al., 2021; Fig.
1a).
Taibei sag, the secondary sag of Turpan depression in Turpan-Hami basin, is the largest sedimentary unit in Turpan-Hami

basin (Fig. 1b). The Taibei sag is a Paleozoic-Cenozoic inherited subsidence area (Li et al., 2021), which is a key area for oil and gas exploration in the Turpan-Hami Basin due to its high thermal evolution degree of hydrocarbon source rocks, good reservoir physical properties, good cap sealing, and rich oil and gas resources, which are the focus of oil and gas exploration in the Turpan-Hami Basin. (Wu et al., 2021; Li et al., 2021). Taodonggou Group is the general name of the Daheyan Formation and the Taerlang Formation. The Daheyan Formation is composed of a sequence of sandstone and conglomerate deposits, with locally interbedded gray to dark gray mudstone. It is unconformably overlain by the Yierxitu Formation. The Taerlang Formation is predominantly composed of gray-black mudstone, with localized occurrences of gray-green siltstone and medium-grained sandstone. Due to the fact that the stratigraphic boundary between the Taerlang Formation and the Daheyan Formation is not obvious, they are collectively called the Taodonggou Group. The Middle Permian Taodonggou Group is mainly located in the western part of the study area. At present, only the YT1 and L30 wells are drilled (the YT1 well is drilled through; the L30 well is not drilled through). The burial depth of the stratum is 4000–6500 m, and the thickness of the mudstone is 50–200 m (Miao et al., 2022b).

**3 Samples and experiments**

**3.1 Samples**

In this study, 16 mudstone samples were collected from well YT1, numbered YT1-1 to YT1-16 in order of depth. After cleaning the samples, XRD, XRF and ICP-MS experiments were conducted.

**3.2 Experiments**

The XRD experiment was carried out at Hangzhou Yanqu Information Co., Ltd. The experimental instrument was the Ultima VI XRD testing instrument of Japanese Neo-Confucianism. In accordance with the Chinese industry standard SY/T 5163-2018, the mudstone was broken to a particle size of less than 200 meshes, and 2 g of samples were weighed to obtain XRD images through Cu/Ka radiation at a scanning speed of 2 °/min. The measurement angle range was $3° \leq 2\theta \leq 70°$, and finally, quantitative interpretation is made with the software X'Pert Highscore Plus of Panalytic Company.

The XRF experiment was conducted in Hangzhou Yanqu Information Co., Ltd., and the experimental instrument was a Panalytical Axios tester from Panalytical. The mudstone was first crushed to a particle size of less than 200 meshes, then 10 g of the sample was weighed and calcined in a muffle furnace for 4 hours to get rid of organic matter and carbonates, weighed and recorded the weight loss, and finally $Li_2B_4O_7$ was added, mixed evenly and made into glass bead, and the main element concentration was tested.

The ICP-MS test was performed at Beijing Orient Smart, and the test instrument was an ELEMENT XR inductively coupled plasma emission spectrometer manufactured by Thermo Fisher, Inc. Before analysis, the samples were ground to a particle size of less than 40 μm. An appropriate amount of the sample was weighed and dissolved in HF (30%) and HNO3 (68%) at 190℃ for 24 hours. After evaporating the excess solvent with deionized water, the solution was redissolved in 2 ml

of 6.5% HNO$_3$. Redissolve in 2 ml of 6 mol/L HNO$_3$ and then store at 150 °C for 48 hours. Subsequently, after evaporating
the solution, 1 ml of the 6 mol/L HNO$_3$ evaporated solution was added to the sample.

## 4 Results

### 4.1 Mineralogy

The XRD test results of 16 samples from Well YT1 are shown in Table A1 and Figure 2. As can be seen from Table A1
and Figure 2, Taodonggou Group mudstones are composed of clay, quartz, calcite, plagioclase, barite, and K-feldspar, and
some samples contain siderite and pyrite. The content of clay is the highest (23.9%–70.9%, mean 40.78%), followed by quartz
(17.2%–59.2%, mean 34.69%), calcite (1%–35.4%, mean 16.97%), barite (0%–13.3%, mean 4.21%), plagioclase (0%–5.4,
mean 2.93%), and K-feldspar (0%–2.3, mean 0.9%).
The mineral composition can be used to analyze the lithofacies type of mudstone, and different lithofacies types often
have different characteristics (Glaser et al., 2014). Previous scholars believed that mudstone types could be divided by the
ternary diagram of mineral composition. The three end elements of the ternary diagram are quartz + feldspar + mica (QFM),
calcite + dolomite + ankerite + siderite + magnesite (carbonate), and clay. The XRD results of 16 mudstone samples from
Well YT1 in the study area are put into the ternary map (Fig. 3). The results show that the data points of Taodonggou Group
mudstone in the study area are located in four areas, namely, mixed mudstone, silica-rich argillaceous mudstone, argillaceous
siliceous mudstone and mixed siliceous mudstone, and most of the points are mixed mudstone and argillaceous siliceous
mudstone areas, which indicates that Taodonggou Group mudstone can be divided into four types: mixed mudstone, silica-
rich argillaceous mudstone, argillaceous siliceous mudstone and mixed siliceous mudstone, and the main lithofacies are mixed
mudstone and argillaceous siliceous mudstone.

### 4.2 Major element

Table A2 shows the results of the major elements in 16 mudstone samples from Well YT1. From Table A2, we can see
that the major elements of the Taodonggou Group mudstone are mainly SiO$_2$, Al$_2$O$_3$, Fe$_2$O$_3$, CaO, and TiO$_2$. The highest
content of SiO$_2$ is from 43.11% to 70.11%, with an average value of 56.18%. Al$_2$O$_3$ content takes second place, accounting
for 11.65% to 25.75%, with an average value of 18.69%; the average content of another main element is less than 10%.

### 4.3 Trace element

The trace element content of the Taodonggou Group mudstone is shown in Table A3. Enrichment factor (EF) is an
important indicator of element enrichment (Taylor and McLennan, 1985; Ross and Bustin, 2009). By comparing the trace
element content of the mudstone of the Taodonggou Group with the global average shale (AS), the trace element enrichment
factors in the study area are calculated as follows:

$$X_{EF} = \frac{(X \, / \, Al)_{samples}}{(X \, / \, Al)_{AS}} \tag{1}$$

Where X and Al represent the concentrations of elements X and Al (Taylor and McLennan, 1985; Ross and Bustin, 2009).
$X_{EF}$ < 1 represents the dilution concentration of element X relative to the standard composition, $X_{EF}$ > 1 represents the relative
enrichment of element X compared to the AS concentration, $X_{EF}$ > 3 represents the detectable autogenetic enrichment, and
$X_{EF}$ > 10 is considered an indicator of moderate to strong autogenetic enrichment (Taylor and McLennan, 1985; Ross and
Bustin, 2009).
Figure 4 and Table A4 presents the enrichment factors of Taodonggou Group mudstone in the study area. It can be seen
from Figure 4 and Table A4 that only Hf (0.5-2.11, mean = 1.29) is enriched in the Taodonggou Group mudstone compared
with AS, and other elements are no enriched.

### 4.4 Rare earth element

The REE content of Taodonggou Group mudstone in the study area is shown in Table A5. According to Table A5, the
∑REE content of Taodonggou Group mudstone ranged from 43.247 ppm to 257.997 ppm, with an average value of 159.206
ppm. The light rare earth element (LREE) content was the highest (mean value 133.45 ppm), followed by medium rare earth
element (MREE) (mean value 17.438 ppm) and heavy rare earth element (HREE) (mean value 6.684 ppm) in that order. After
chondrite standardization (Taylor and Mclennan, 1985), Taodonggou Group mudstone shows a right dipping REE distribution
pattern (Fig. 5), $(La/Yb)_N$ is 6.228–10.081, with an average value of 7.358.
In addition, in Figure 5, although the YT1-13 sample exhibits a weak right dipping REE distribution pattern similar to
other samples, its rare earth elements are significantly depleted. Based on Figure 4 and Table A4, the trace elements in the
YT1-13 sample are depleted compared to AS, indicating that the YT1-13 sample has been influenced by groundwater leaching.

### 4.5 Reconstruction of paleosedimentary environment based on element geochemical characteristics

4.5.1 Paleoclimate and weathering
The paleoclimate not only affects the weathering degree of the parent rock but also affects the transport distance of
sedimentary debris and the transport of nutrients (Zhang et al., 2005). There are many evaluation indices for paleoclimate,
such as the chemical alteration index (CIA) and the climate index (*C*). It is generally believed that when CIA=50–65 and *C* <
0.2, it reflects that the sedimentary system is in a dry and cold climate under the background of lower of degree of chemical
weathering; when CIA=65–85 and 0.2 < *C* < 0.8, it indicates that the sedimentary system is in a warm and humid climate
under the background of middle of degree of chemical weathering; when CIA=85–100 and *C* > 0.8, it reflects the humid and
hot climate under the background of high of degree of chemical weathering (Zhang et al., 2019; Nesbitt and Young, 1984).
The calculation formula for CIA and C is as follows:
$$CIA = \frac{Al_2O_3 \times 100}{Al_2O_3 + Na_2O + CaO* + K_2O} \qquad (2)$$

$$C = \frac{Fe + Mn + Cr + Ni + V + Co}{Ca + Mg + Sr + Ba + K + Na} \qquad (3)$$

In formula (2), CaO * only refers to CaO in silicate minerals. Due to the lack of direct measurement means, it is often
calculated indirectly by the content of $P_2O_5$, namely:
$$CaO* = mol(CaO) - \frac{10}{3}mol(P_2O_5) \qquad\qquad\qquad\qquad (4)$$
Where, mol(CaO) and mol($P_2O_5$) are the mole numbers of CaO and $P_2O_5$, where when mol ($Na_2O$) ≤ mol (Cao *), mol
(Ca0 *) = mol ($Na_2O$); on the contrary, when mol($Na_2O$) > mol(CaO*), mol(CaO*)=mol(CaO) (Nesbitt and Young, 1984).
The CIA values of the Taodonggou Group mudstone in the study area were calculated based on Equation (2) and Equation
(3), ranging from 68.71 to 96.97, with a mean value of 80.17. The climate index (*C*) is 0.22–2.42 (average = 1.01, Tab.3). The
overall paleoclimate was warm, humid, and hot (Fig. 6a).
In addition, the cross plot of Ga/Rb and $K_2O/Al_2O_3$ can also be used to analyze the paleoclimate characteristics during
the formation of sedimentary rocks (Lerman and Baccini, 1987; Liu and Zhou, 2007). As shown in the cross plot of Ga/Rb
and $K_2O/Al_2O_3$ (Fig. 6b), almost all points are in the warm/wet area, which indicates that Taodonggou Group mudstone was
deposited in a warm and humid paleoclimate.
By analyzing the correlations between CIA, *C*, and Ga/Rb (Figure 6c–e), it can be observed that there is the strongest
correlation between CIA and *C* (Figure 6c, $R^2$ = 0.7566). Additionally, the correlation coefficients between CIA and Ga/Rb,
as well as *C* and Ga/Rb, are both greater than 0.4 (Figures 6d and 6e). This indicates that CIA, *C*, and Ga/Rb are reliable
indicators of the paleoclimate during the sedimentation of the Taodonggou Group mudstone. Based on the above analysis, the
Taodonggou Group mudstone in the study area was deposited in a warm, humid, and hot paleoclimate. This result is consistent
with Miao's indicator result using the biomarker parameter CPI (Miao et al., 2021), indicating that the biomarker parameter
CPI can be used to explain the paleoclimate change characteristics of hydrocarbon source rocks with Ro ≤ 1.49.
4.5.2 Paleo-redox conditions
Redox environments are critical to the preservation of organic matter in sedimentary rocks, and sensitive elements such
as Co, Mo, U, Th, V, Ni, and Cr are commonly used to identify redox conditions in ancient water bodies. Previous evidence
suggests that U/Th < 0.75, V/Cr < 2 and V/(V+Ni) < 0.45 represent an oxic conditions, 0.75 < U/Th < 1.25, 2 < V/Cr < 4.25
and 0.45 < V/(V+Ni) < 0.84 represent a dyoxic conditions, U/Th < 1.25, V/Cr < 4.25 or V/(V+Ni) < 0.84 represent an anoxic
condition (Hatch and Leventhal, 1992; Rosenthal et al., 1995; Tribovillard et al, 2006; Tribovillard et al, 2012). There is no
significant correlation between V, U, and Th and $Al_2O_3$ contents in the Taodonggou Group mudstone samples, indicating that
V, U, and Th contents in Taodonggou Group mudstone are mainly controlled by authigenic deposition under redox conditions
(Tribovillard et al., 1994). The U/Th, V/Cr, and V/(V+Ni) of the Taodonggou Group mudstone range from 0.21 to 0.52 (mean
= 0.29), 1.62 to 4.95 (mean = 2.7), and 0.65 to 0.92 (mean = 0.75). In the light of U/Th, Taodonggou Group mudstones were
deposited in an oxic environment, and according to V/Cr and V/(V+Ni), Taodong Group mudstones were deposited in a
dyoxic environment. This is because U/Th cannot accurately identify the redox environment of the sediments under highly
weathered conditions (Cao et al., 2021), so V/Cr and V/(V+Ni) were used in this study to identify the redox environment of
Taodonggou Group mudstone. The cross plot of V/Cr and V/(V+Ni) shows (Fig. 7) that Taodonggou Group mudstones were
deposited in a dyoxic environment.
4.5.3 Paleosalinity
Paleosalinity is an important indicator of the paleoenvironment of a water body. The level of paleosalinity affects the
stratification of the sedimentary water body and the development of plankton, thereby affecting the paleoproductivity and
enrichment of organic matter in the sedimentary environment (Thorpe et al., 2012; Wang et al., 2021; Shi et al., 2021).
Previous research has found that Sr/Ba and B/Ga can represent changes in paleosalinity. It is generally believed that Sr/Ba<0.5
or B/Ga<3 represents fresh water, 0.5<Sr/Ba<1 or 3<B/Ga<6 means brackish water, and Sr/Ba>1 or B/Ga>6 represents saline
water. The correlation between Sr and CaO of Taodonggou Group mudstone in the study area is not obvious ($R^2$=0.17), Sr/Ba
of Taodonggou Group mudstone in the study area ranges from 0.32 to 1.83, with an average value of 0.71, and the B/Ga is
2.53–5.81 (average = 3.36), indicating that Taodonggou Group mudstone was deposited in freshwater and brackish water
environments (Fig. 8a).
In addition, Ca/(Ca+Fe) is a reliable indicator for evaluating the salinity of lake waters (Wang et al., 2021). The
Ca/(Ca+Fe) distribution of Taodonggou Group mudstone in the study area ranges from 0.14 to 0.78, with a mean value of
0.42. The Sr/Ba and Ca/(Ca+Fe) intersection diagram (Fig. 8 b) shows that Taodonggou Group mudstones were deposited in
freshwater and brackish water environments, which is in accord with the Sr/Ba and B/Ga intersection diagram.
4.5.4 Paleobathymetry
Previous research has shown that some elements of the sedimentation process change dramatically with offshore distance.
These elements can be used to judge the water depth variation during the sedimentation period. The commonly used indicators
are Zr/Al, Rb/K, and MnO content (Xiong and Xiao, 2011; Herkat et al., 2013). It is now believed that the lower the Zr/Al
ratio or the higher the Rb/K ratio, the further offshore and the deeper the water (Xiong and Xiao, 2011; Herkat et al., 2013).
Zr/Al of Taodonggou Group mudstone is $5.19 \times 10^{-4}$– $22.51 \times 10^{-4}$ (average = $13.44 \times 10^{-4}$), showing a trend of first decreasing
and then increasing with the depth, Rb/K ranges from $7.32 \times 10^{-4}$ to $29.79 \times 10^{-4}$ (mean $19.02 \times 10^{-4}$), with large fluctuations
with depth of burial. The high-value area of Rb/K is basically consistent with the low-value area of Zr/Al, which indicates
that the ancient water depth during the Taodonggou Group mudstone deposition process has a trend of first decreasing and
then increasing.
For the content of MnO, it is generally believed that < 0.00094% is a shore lake, 0.00094%–0.0075% is a shallow lake,
0.0075%–0.051% is an intermediate-depth lake, and > 0.051% is a deep lake (Herkat et al., 2013). MnO of Taodonggou Group
mudstone is 0.05%–0.30%, with an average of 0.16 %, which indicates that the Taodonggou Group mudstone are mainly
deposited in intermediate depth - deep lake sedimentary environment.
4.5.5 Terrigenous detritus input
Ti, Si, and Al are relatively stable during diagenesis and are usually used as indicators of debris flux input (Algeo and

Maynard, 2004; Maravelis et al., 2021). Generally, Ti in sediments comes from ilmenite ($FeTiO_3$) or rutile ($TiO_2$), while Al

can exist in feldspar, clay minerals, and other aluminum silicate minerals (Algeo and Maynard, 2004). Compared with Ti and

Al, Si comes from many sources, including both biological origin and hydrothermal and terrigenous clastic input (Kidder and

Erwin, 2001). Therefore, when using $SiO_2$ as the evaluation index for terrigenous clastic input, its source needs to be analyzed.

The correlation of $Al_2O_3$ and $TiO_2$ with $SiO_2$ in Well YT1 of the study area is not obvious, which indicates that their sources

are more complex and not dominated by terrestrial debris sources (Fig. 9). Therefore, $Al_2O_3$ and $TiO_2$ are used in this study

to indicate the terrestrial debris input during the deposition of the Taodonggou Group mudstone.

The $Al_2O_3$ content of YT1 wells is higher, ranging from 11.65 % to 25.75 %, with an average value of 18.69 %; the $TiO_2$

is 1.15 %–4.22 % (average = 1.77 %). As can be seen from Table A2, the $Al_2O_3$ content of Well YT1 fluctuates more with

depth, and the overall trend is increasing first and then decreasing with depth, while the $TiO_2$ fluctuates less with depth, and

on the whole, the trend is increasing with depth. Combined with the results of paleoclimate analysis in the study area, it is

found that the terrestrial debris input during the deposition of the Taodonggou Group strata has the characteristics of increasing

first and then decreasing.

4.5.6 Paleoproductivity

Paleoproductivity determines the quantity of original organic matter in sedimentary rocks (Wei et al., 2012; Algeo and

Ingall, 2007; Ross and Bustin, 2009; Schoepfer et al., 2015). The elements P, Si, Ba, Zn, and Cu are indicators of the magnitude

of paleoproductivity, but they all have a certain range of application; for example, only the biogenic parts of Si and Ba can

represent productivity, and Zn can only represent productivity change in the sulfide reduction environment (Wei et al., 2012;

Algeo and Ingall, 2007).

P is not only a key nutrient element in biological metabolism but also an important component of many organisms, so it

can also be used to characterize biological productivity (Kidder and Erwin, 2001). P/Ti or P/Al is commonly used to reflect

biological productivity in order to eliminate the influence of terrigenous detritus. The P/Ti of Taodonggou Group mudstone

in the study area ranges from 0.04 to 0.74 percent, with an average value of 0.17 percent and an overall low productivity. As

shown in Table A2, the relationship between P/Ti and depth was analyzed, and the results showed that the paleontological

productivity tended to increase and then decrease with depth.

In addition, Cu is also an important nutrient and, unlike P, is generally indicative of productivity, including the sum of

primary productivity and productivity from terrestrial inputs (Schoepfer et al., 2015). For the purpose of eliminating the

dilution interference of terrigenous detritus, Cu/Ti is used as an indicator to evaluate the paleoproductivity in this study. The

distribution range of Cu/Ti of Taodonggou Group mudstone in the study area is from 0.55 to 1.96 with an average value of

1.02 and gradually decreases with depth, indicating a gradual increase in palaeoproductivity during the deposition of

Taodonggou Group mudstone.

4.5.7 Deposition rate

The deposition rate is one of the parameters characterizing the magnitude of the dilution effect during deposition and is commonly characterized by $(La/Yb)_N$. It is generally believed that the difference between LREE and HREE migration is not significant when the sedimentation rate of the lake basin is faster and the $(La/Yb)_N$ value is close to 1. Conversely, when the $(La/Yb)_N$ value is greater or less than 1, it indicates that the sedimentation rate of the lake basin is slower (Wang et al., 2021; Cao et al., 2018). The $(La/Yb)_N$ of the Taodonggou Group mudstones are 6.228–10.081, with an average value of 7.358 in the study area, which is much greater than 1. This indicates that the mudstone of the Taodonggou Group has a slower deposition rate.

4.5.8 Hydrothermal activity

The study area has been extremely volcanically active from the Carboniferous to the Permian, with extensive volcanic deposits in the Middle Permian Taodongou Group, the Lower Permian Yierxitu Formation, and the Carboniferous. In order to explore whether hydrothermal activity is involved in the Middle Permian sedimentation, the Zn-Ni-Co ternary diagram and the $(Cu+Co+Ni)\times10$-Fe-Mn ternary diagram are applied in this study (Xu et al., 2022; You et al., 2019). Based on the Zn-Ni-Co ternary diagram (Fig. 10a), some data points of the Taodonggou Group mudstone are distributed in the hydrothermal sedimentary zone, and based on the $(Cu+Co+Ni)\times10$-Fe-Mn ternary diagram (Fig. 10b), all data points of the samples fall in the hydrothermal sediment zone and Red Sea hydrothermal sediment zone, which indicates that the Taodonggou Group mudstone deposition was influenced by hydrothermal fluids.

4.5.9 Tectonic setting

Sedimentary rocks of different tectonic settings have prominent differences in element composition and content, so the geochemical characteristics of sedimentary rocks can be used to reflect the tectonic setting of sedimentary basins (Kroonenberg, 1992).

The elements Co, Th, Sc, Zr, and La are relatively stable and less affected by geological activities such as weathering, transportation, and deposition. Therefore, the La-Th-Sc ternary diagram and the Th-Co-Zr/10 ternary diagram can be utilized to distinguish the tectonic setting during the formation of sediments (Bhatia and Crook, 1986; Cai et al., 2022). Based on the La-Th-Sc ternary diagram (Fig. 11a), most of the data points fall in the continental island arc region, and on the Th-Co-Zr/10 ternary diagram (Fig. 11b), almost all the data points fall in the continental island arc and oceanic island arc regions. This indicates that the tectonic setting of the Taodonggou Group's source area is a continental island arc and an oceanic island arc.

Additionally, previous studies have shown that $SiO_2$, $TiO_2$, $Al_2O_3/SiO_2$ and $Fe_2O_3+MgO$ are also important parameters for identifying the source tectonic setting. Cross plots of $Al_2O_3/SiO_2$ and $Fe_2O_3+MgO$, $TiO_2$ and $Fe_2O_3+MgO$, and $SiO_2$ and $Al_2O_3/SiO_2$ are often employed to recognize the tectonic setting (Bhatia, 1983; Li et al., 2020; Roser and Korsch, 1988). Based on the cross plot of $Al_2O_3/SiO_2$ and $Fe_2O_3+MgO$ (Fig. 11c), all data points are distributed around the continental island arc

and oceanic island arc, which is consistent with the cross plot of $TiO_2$ and $Fe_2O_3+MgO$ (Fig. 11d) and the cross plot of $SiO_2$ and $Al_2O_3/SiO_2$ (Fig. 11e). As a result, the tectonic setting of Taodonggou Group mudstone source area is continental island arc and oceanic island arc.

## 5 Discussion

The sedimentary environment, provenance location, and sedimentation mode are factors that influence the quality of mudstones. In this study, based on the mineralogical and elemental geochemical characteristics of the Taodonggou Group mudstones, we discuss the influence of sedimentary environment, provenance location, and sedimentation mode on the quality of the Taodonggou Group mudstones.

### 5.1 The influence of palaeosedimentary environment on the quality of mudstone

Based on the mineralogical, elemental geochemical characteristics and previous studies on the organic geochemical characteristics of the Taodonggou Group mudstones (Miao et al., 2021), a comprehensive geochemical profile of the YT1 well was established. The results are shown in Figure 12. It can be observed from Figure 12 that the sedimentary environment of the Taodonggou Group mudstones is closely related to their organic matter types and can be divided into three periods. In the early stage of the Taodonggou Group, the overall climate was warm and humid under moderate chemical weathering conditions. The sedimentary water body was dyoxic-anoxic brackish water. At this time, productivity was weak, and organic matter was mainly derived from terrestrial sources. In the middle stage of the Taodonggou Group, the paleoclimate gradually shifted to a dry and humid climate under strong chemical weathering conditions, accompanied by hydrothermal activity. This provided abundant nutrients for the growth of algae and other microorganisms. At the same time, the sedimentation rate increased, resulting in a predominance of algae in the organic matter composition during this period. During the late stage of the Taodonggou Group, the climate again shifted to a warm and humid climate under moderate chemical weathering conditions. The sedimentation rate slowed down, and the input of organic matter shifted back to predominantly terrestrial sources.

### 5.2 Provenance

5.2.1 Lithology of parent rock

Previous studies have found that the chemical composition of the rocks in the sedimentary area and the parent rock in the provenance area have a strong affinity, and the type of parent rock will directly affect the elemental geochemical characteristics of the sediment (Tribovillard et al., 2006; Shi et al., 2021; McLennan et al., 1993; Basu et al., 2016; Hu et al., 2021; Floyd and Leveridge, 1987; Wronkiewicz and Condie, 1987). Generally speaking, the transport of sediment from the source area to the sedimentary area goes through multiple complex processes such as mechanical transport and chemical action, and hence it is necessary to analyze the impact of sediment sorting and recycling on each chemical component when identifying the source. Previous studies have shown that trace elements Zr, Th, and Sc are relatively stable in geological

processes such as weathering, transportation, and sorting and are not easily lost, which can be used as one of the indicators
for parent rock identification (Floyd and Leveridge, 1987; Wronkiewicz and Condie, 1987). According to the Th/Sc and Zr/Sc
intersection diagram of Taodonggou Group mudstone (Fig. 13a), Taodonggou Group mudstone is close to andesite and felsic
volcanic rock of the upper crust, and its composition is controlled by the composition of its felsic parent rock and has not
undergone sediment sorting and recycling.
In addition, REE and trace elements in mudstone from different parent rocks are obviously different, so the ratio of REE
to trace elements can be used to analyze the type of parent rock, and the most common ones are La/Sc, La/Co, Th/Sc, Th/Co,
and Cr/Th (Basu et al., 2016; Hu et al., 2021; Floyd and Leveridge, 1987; Wronkiewicz and Condie, 1987; Allègre and Minster,
1978). Based on the Hf and La/Th intersection diagrams (Fig. 13b) and the La/Sc and Co/Th intersection diagrams (Fig. 13c),
we can see that the mudstones of the Taodonggou Group have both andesitic island-arc sources and felsic volcanic sources.
It can be seen from the cross plot of $TiO_2$ and Zr (Fig. 13d) that the mudstone of the Taodonggou Group is a source of
intermediate igneous rocks and felsic igneous rocks. As can be seen from the cross plot of La/Yb and Σ REE (Fig. 13e), almost
all data points are located in the sedimentary rock, alkali basalt, and granite areas.
In summary, the parent rocks of the Taodonggou Group mudstone are andesitic and feldspathic volcanic rocks with weak
sedimentary sorting and recirculation, and the material source information is well preserved.
5.2.2 Location of Parent Rock
There is a great deal of controversy about the provenance location of the Middle Permian in Turpan-Hami (Shao et al.,
2001; Jiang et al., 2015; Wang et al., 2019; Zhao et al., 2020; Song et al., 2018; Wang et al., 2018; Tang et al., 2014). Shao et
al. (1999) believed that the provenance of the Permian was mainly from the Jueluotage Mountain in the south of the Turpan-
Hami Basin; Song et al. (2018) considered that it came from the Bogda area; Zhao et al. (2020) believed that the provenance
of the Permian in the Turpan-Hami Basin was consistent with that in Junggar and originated from the Kelameili Mountain
and the Northern Tianshan. Summarizing the previous research results, it is found that the main controversial point is the time
of the first uplift of Bogda Mountain.
At present, there are many opinions about the time of the Bogda Mountain uplift. They think that the initial uplift of
Bogda Mountains occurred in Early Permian (Carroll et al., 1990; Shu et al., 2011; Wang et al., 2018; Li et al., 2022), Middle
Permian (Zhang et al., 2006; Liu et al., 2018; Wang et al., 2018), Late Permian-Early Triassic (Zhao et al., 2020; Guo et al.,
2006; Wang, 1996; Sun and Liu, 2009; Tang et al., 2014; Wang et al., 2018), Middle Triassic (Guo et al., 2006), Early Jurassic
(Green et al., 2005; Liu et al., 2017; Ji et al., 2018) and Late Jurassic (Yang et al., 2015). If the initial uplift of the Bogda
Mountains was after the middle Permian, the parent rock types of the Taodonggou Group mudstone in the Turpan-Hami Basin
and the Luchaogou Formation mudstone in the Junggar Basin should be the same.
We have counted the element geochemical characteristics of Luchaogou Formation in the Junggar Basin (Li et al., 2020)

and found that the parent rock type of Luchaogou Formation mudstone in the Junggar Basin is greatly different from that of P$_2$td, which is felsic volcanic rock (Fig. 14). As a result, Bogda Mountain's initial uplift should be Late Permian-Early Triassic in the Early Permian or Middle Permian. This is consistent with Li et al. (2022) and Wang et al. (2018), who inferred the uplift of Bogda Mountain at 289.8 Ma–265.7 Ma. Shao et al. (2001) believed that the sandstone of the Daheyan Formation in Turpan-Hami Basin has a good affinity with the Early Permian and Carboniferous, so the provenance direction of the sandstone of the Daheyan Formation is consistent with that of the Early Permian, and they all come from the Jueluotage Mountain. However, the paleocurrent direction of the Early Permian in Xinjiang is southeast (Zhang et al., 2005; Li et al., 2007; Wang et al., 2019), and the provenance area is located in the north of the Bogda area. Zhao et al. (2020) calculated the U-Pb dating results of 5250 zircons in the Tianshan and believed that the provenance of the Turpan-Hami Basin and the Junggar Basin both came from the northern Tianshan and the Kelameili Mountain, which is also consistent with the ancient ocean current direction in the Early Permian (Zhang et al., 2005; Li et al., 2007; Wang et al., 2019; Fig. 14a). Consequently, the first uplift of Bogda Mountain should have occurred in the early Permian, but it was not exposed in the early Middle Permian, and it still received sedimentation. In the middle Permian, the exposed water began to be denuded, becoming the source area of the Turpan-Hami Basin (Wang et al., 2018).

Based on the above analysis, in the early Middle Permian, although Bogda Mountain in the north of Turpan-Hami Basin was uplifted due to orogeny, it did not emerge from the water surface, and it still accepted the provenance of North Tianshan and Kelameili Mountain. At this time, there was a NE-trending ancient ocean current (Carrollet et al., 1995; Obrist-Farnert et al., 2015; Zhao et al., 2020), so Jueluotage Mountain, which has been uplifted in the south of Turpan-Hami Basin, became a secondary provenance area (Shao et al., 1999; Fig. 14b). With the continuous uplift of Bogda Mountain, the sedimentary center of Turpan-Hami Basin gradually shifted to Taibei Sag, and the provenance area of Turpan-Hami Basin changed to Bogda Mountain and Jueluotage Mountain (Fig. 14c).

**5.3 sedimentation mode**

In previous studies, scholars have believed that the sedimentation of the Permian in the Turpan-Hami Basin is mainly controlled by traction currents (Chen et al., 2003). However, recent research has revealed the presence of gravity flow deposits in the Permian of the Turpan-Hami Basin (Wang et al., 2017; Wang et al., 2018; Xu, 2022). Yang et al. (2010) found poorly sorted debris flow deposits in the Daheyan Formation, and Xu (2022) discovered alluvial and fluvial facies in the Daheyan Formation, consisting of volcaniclastic rocks and conglomerates that are similar in composition to the Lower Permian volcaniclastic rocks and conglomerates. This suggests the existence of gravity flow deposits during the early Permian in the Turpan-Hami Basin. Wang et al. (2018) also suggested the development of gravity flow deposits and pillow lavas in the Early Permian. Meanwhile, in the early Middle Permian, the sedimentation inherited the provenance and sedimentation style from the early Permian, but the gravity flow deposits transitioned gradually into traction current deposits. Due to the influence of

gravity flow deposits, terrestrial organic matter can be transported to the deep lake area (Yu et al., 2022; Li et al., 2011), thereby altering the type of organic matter.

During the middle of the Taodonggou Group, the Turpan-Hami Basin entered the foreland basin sedimentation stage due to the uplift of the Bogda Mountains. The sedimentary environment of the Taodonggou Group in the Tainan Sag is similar to that in the Taibei Sag (Li, 2019). During this time, the sedimentary water body of the Taodonggou Group in the Turpan-Hami Basin became shallower, and the dominant sedimentation style transitioned to traction currents. Xu (2022) conducted lithological observations on the Taerlanggou section, the Zhaobishan section, and the Y well in the Taodonggou Group and found the presence of traction structures of gravity flow origin in the middle and upper parts of the Taerlang Formation. Additionally, a large number of calcareous and iron nodules appeared in the formation, indicating the occurrence of gravity flow deposits during the late-stage sedimentation of the Taodonggou Group. The organic matter type in the mudstones during this period was influenced by gravity flows.

**5.4 Formation mechanism of the Taodonggou Group mudstone**

Based on the sedimentary environment, provenance, and sedimentation mode during the deposition of the Taodonggou Group mudstones, this study has constructed the formation mechanism of the Taodonggou mudstones. The results indicate that the formation of the Taodonggou Group mudstones can be divided into three stages.

In the early of the Taodonggou Group, Bogda Mountain began to rise but did not emerge from the water surface. The sediment source is mainly from North Tianshan and Kelameili Mountain, and the secondary source area is Jueluotage Mountain in the south of the Turpan-Hami basin. The stratum of the Taodonggou Group was deposited in a warm and humid paleoclimate with high weathering intensity and a stable input of terrigenous detritus. In addition, the sedimentary water body is deep at this time, creating a deep lake environment of brackish water and dyoxic. However, this period inherited the gravity flow sedimentation characteristics from the Early Permian. Due to the influence of gravity flows, terrestrial organic matter was transported to the deep lake, resulting in the input of organic matter in the mudstones primarily derived from terrestrial higher plants (Miao et al., 2021). Consequently, a high-quality Type III organic matter source rock was formed (Fig.15a).

In the middle of the Taodonggou Group, with the continuous uplift of Bogda Mountain and hydrothermal activity, the climate changed into a hot and humid paleoclimate, the weathering degree further increased, and the input of terrigenous detritus increased. The provenance areas are Bogda Mountain and Jueluotage Mountain. In addition, during this period, the sedimentary center gradually transferred to the Taibei sag, and the sedimentary water body became shallow, which was a dyoxic intermediate-depth lake environment. Due to the nutrients brought by hydrothermal activities, the lower algae multiplied during this period, and the salinity of the sedimentary water body became lower, becoming a freshwater environment and thus depositing a set of high-quality $II_2$ organic source rocks.

In the late Taodonggou Group, the uplift of Bogda Mountain basically stopped, and the climate changed to a warm and humid paleoclimate again. The weathering degree was high, and the input of terrigenous debris was reduced. Bogda Mountain

and Jueluotage Mountain remained the provenance areas. The sedimentary center was essentially transferred to the Taibei Sag at this time. During this period, the salinity of the sedimentary water body was high, and the sedimentary water body became deeper. It was a deep lake environment with dyoxic and brackish water. During this period, the sedimentation was also influenced by gravity flows, leading to changes in lithology and organic matter type. As a result, the organic matter type in the mudstones deposited during this period transitioned to Type III.

## 6 Conclusion

Through the mineral composition and element geochemistry analysis of the Taodonggou Group mudstone, the following understandings have been obtained:

(1) The mudstone minerals of the Taodonggou Group are mainly clay and quartz and can be classified into 4 petrographic types according to their mineral fractions.

(2) The Taodonggou Group mudstone was deposited in an intermediate-depth or deep, dyoxic, freshwater-brackish lake environment under warm and humid paleoclimatic conditions. The input of terrestrial debris was stable, but the sedimentation rate was slow. In addition, the sedimentation in the middle stage was influenced by hydrothermal activities. In addition, the source rocks of the Taodonggou Group mudstone are mainly andesitic and feldspathic volcanic rocks. Sediment sorting and recycling were weak, and hydrocarbon source information was well preserved. The tectonic background of the source area was a continental island arc and an oceanic island arc.

(3) The sedimentary environment, sources, and sedimentary methods have significant impacts on the organic matter types of the Taodonggou Group. In the early taodonggou Group, the sedimentation center was in the Bogda area. At this time, the Bogda Mountain region was not exposed, and the depositional processes inherited the characteristics of Early Permian gravity flow sedimentation, resulting in the widespread deposition of a series of high-quality Type III source rocks in the basin. In the middle taodonggou Group, the sedimentation center gradually migrated to the Taibei Sag. During this period, the Bogda Mountain region experienced uplift and hydrothermal activity, and the depositional processes gradually transitioned to traction flows, resulting in the widespread deposition of a series of Type II source rocks in the basin. In the late taodonggou Group, the uplift of the Bogda Mountain region ceased, and the sedimentation center completely shifted to the Taibei Sag. Meanwhile, under the influence of gravity flows, the organic matter types of the Taodonggou mudstone changed to Type III.

**Data availability**

Data will be made available on request.

**Acknowledgement**

This study was supported by National Major Science and Technology Project of China (grant nos. 2016ZX05066001-002; 2017ZX05064-003-001; 2017ZX05035-02 and 2016ZX05034-001-05), Innovative Research Group Project of the National Natural Science Foundation of China (grant nos. 41872135 and 42072151), PetroChina Science and Technology

Project (grant nos. 2021DJ0602) and National Energy Shale Gas R & D (Experiment) Center (Grant nos. 2022-KFKT-15). We thank Hangzhou Yanqu Information Co., Ltd, Key Laboratory of natural gas accumulation, China National Petroleum Corporation and development and Beijing Orient Smart for providing testing samples and test equipments, as well as our colleagues' useful suggestions.

**Author contribution**

Miao H. and Guo J.Y. designed experiments, Wang Y.B. and Jiang Z.X. revised the first draft of the manuscript, Guo J. Y., Wang Y.B. and Jiang Z.X. provided financial support, Miao H. and Zhang C.J. provided language services and figure production, Li C.M. investigated and revised the ideas of the article, and Miao H. prepared the manuscript with your contributions. All authors contributed to the review of the manuscript.

**Competing interests**

The contact author has declared that none of the authors has any competing interests.

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

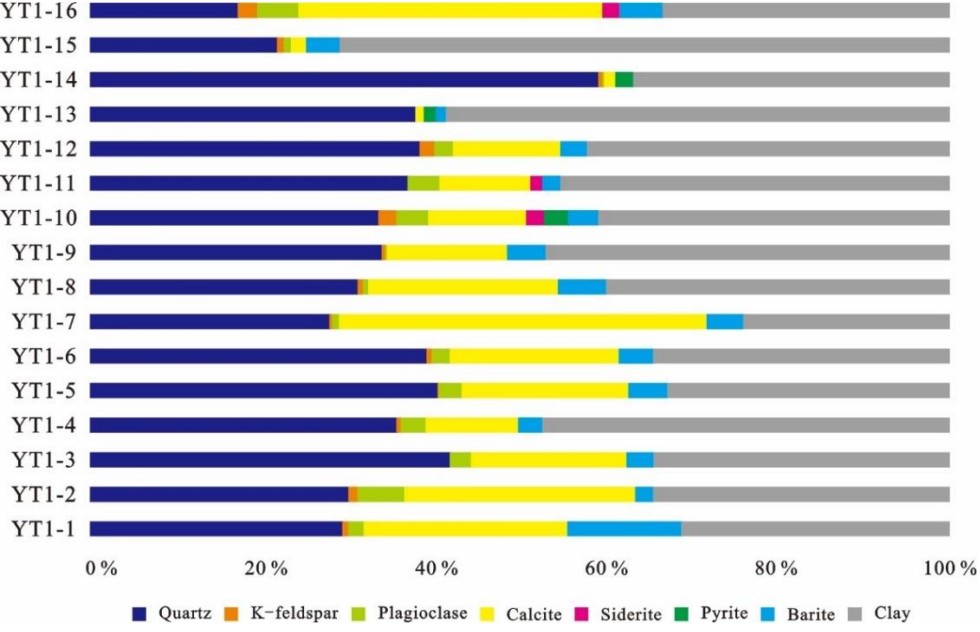


**Figure1: Geological overview of the study area (modified after Miao et al., 2021; Miao et al., 2023): (a) Geological background of Turpan-**
**Hami basin; (b) Thickness contour map of Taodonggou Group mudstone in Taibei sag; (c) YT1 stratum of Taodonggou Group**

**Figure 2: Mineral composition of Taodonggou group mudstone in YT1 well**

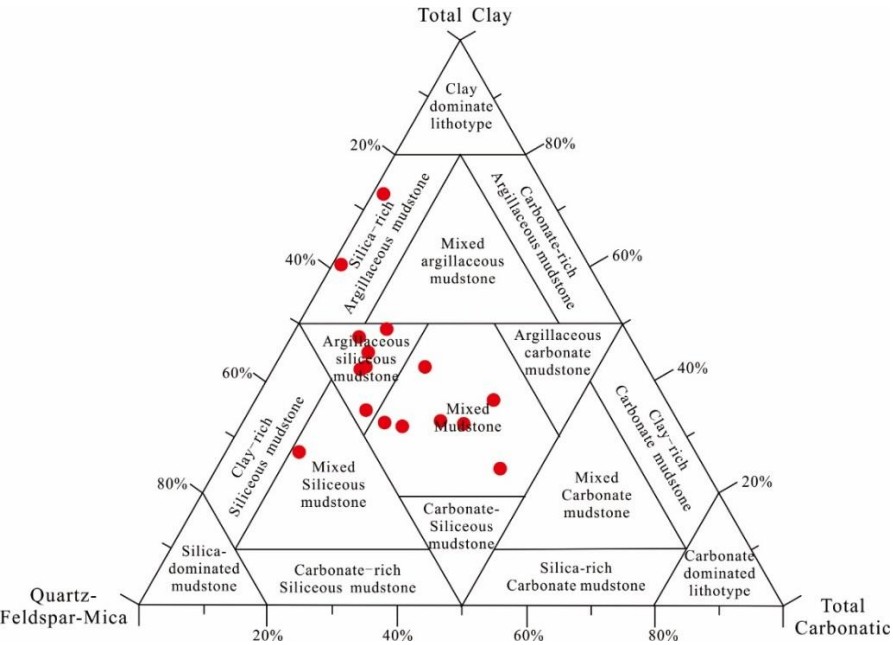


**Figure 3: Lithofacies classification of Taodonggou Group mudstone in well YT1(modified from Glaser et al., 2014)**

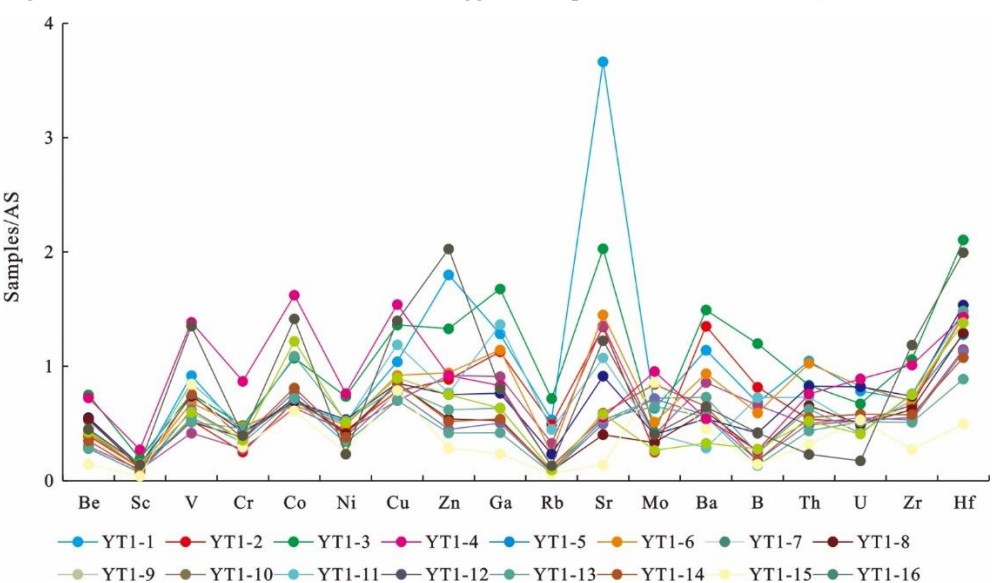


**Figure 4: AS standardized multi-element diagrams of Taodonggou Group mudstone in the study area.**

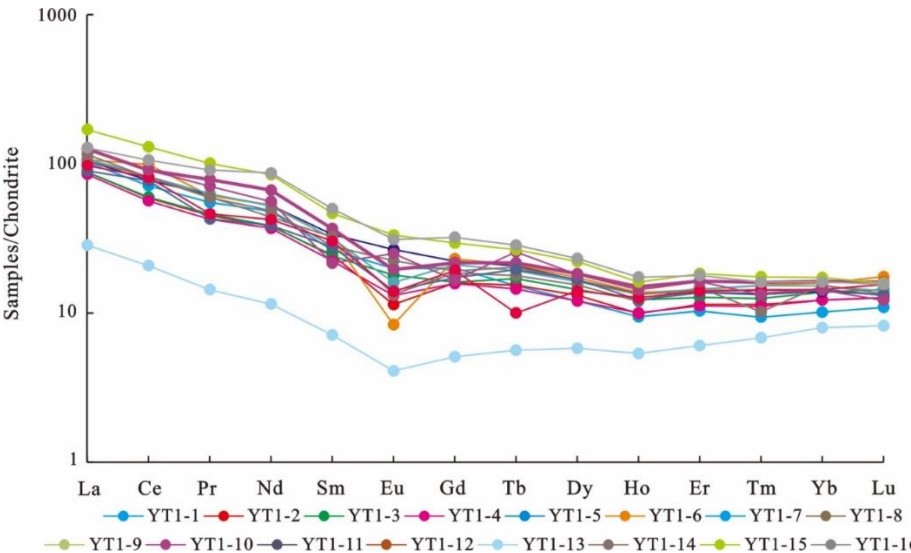


**Figure 5: Standardized map of rare-earth element chondrite in mudstone of Taodonggou Group**

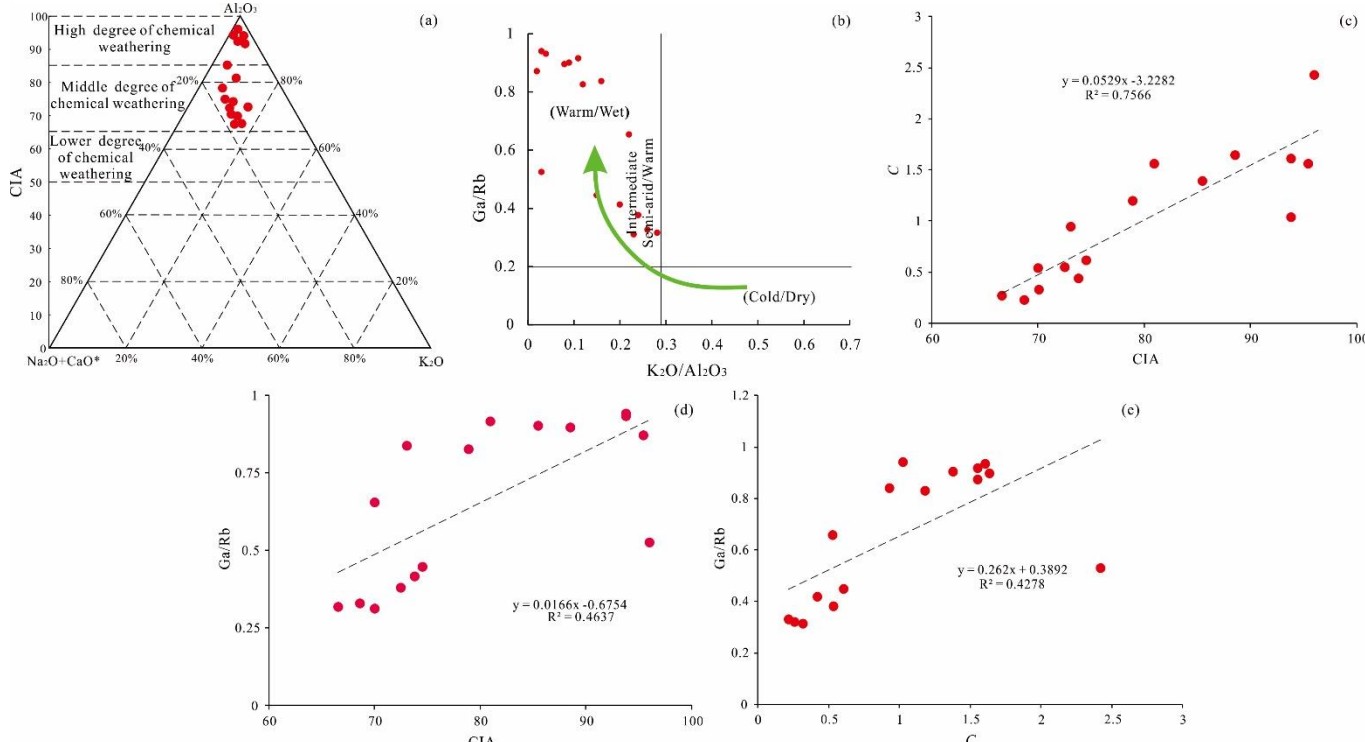


**Figure 6: Paleoclimate of Taodonggou Group: (a) CIA Characteristics of Taodonggou Group mudstone (modified from Nesbitt and Young,**
**1984); (b) cross plot of $K_2O/Al_2O_3$ and Ga/Rb (modified from Roy and Roser, 2013); (c) cross plot of CIA and $C$; (d) cross plot of CIA and**
**Ga/Rb;(e)cross plot of $C$ and Ga/Rb**

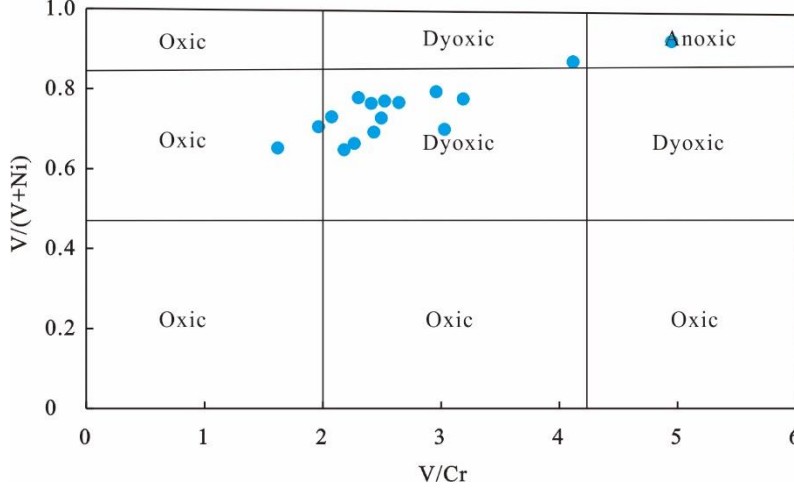


**Figure7: Cross plot of V/Cr and V/(V+Ni)**

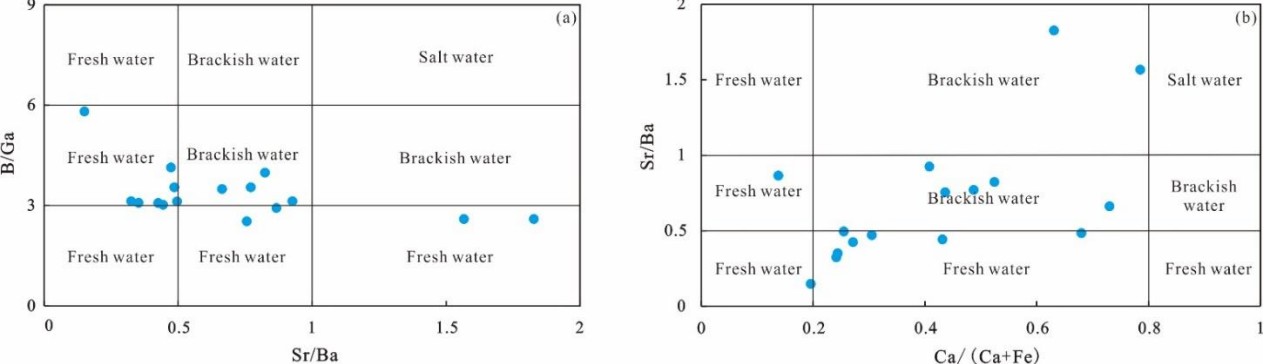


**Figure 8: Cross plot of B/Ga and Sr/Ba (a) and cross plot of Ca/(Ca+Fe) and Sr/Ba (b)**

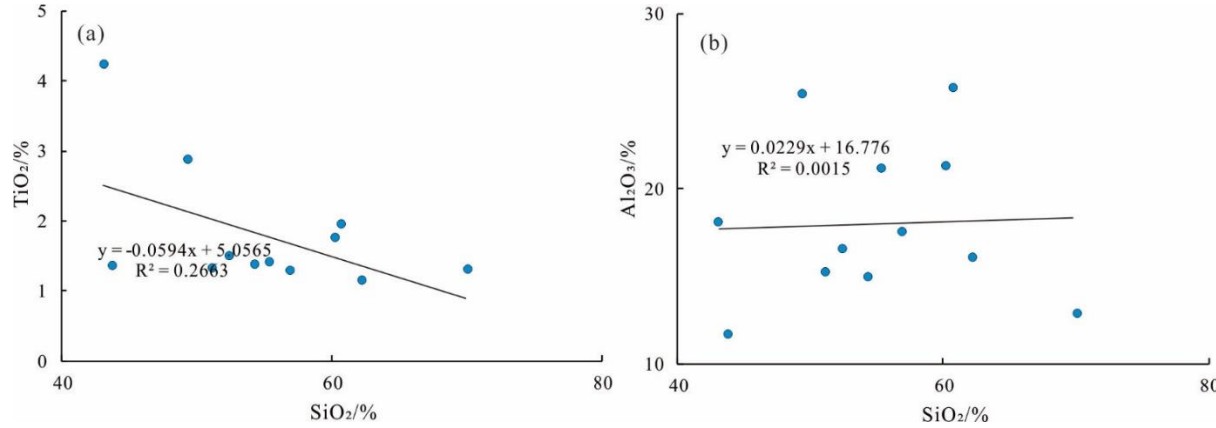


**Figure 9: Intersection diagram of TiO₂ and SiO₂ (a) and intersection diagram of Al₂O₃ and SiO₂ (b)**


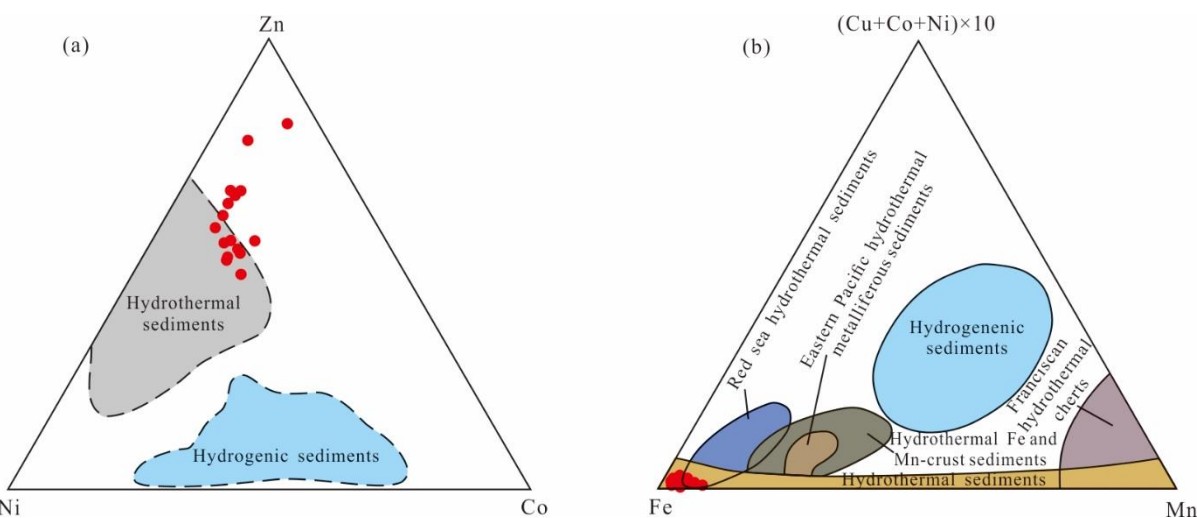


**Figure 10: Zn-Ni-Co ternary diagram (a) and (Cu+Co+Ni) ×10-Fe-Mn ternary diagram (b) (modified after You et al., 2019)**


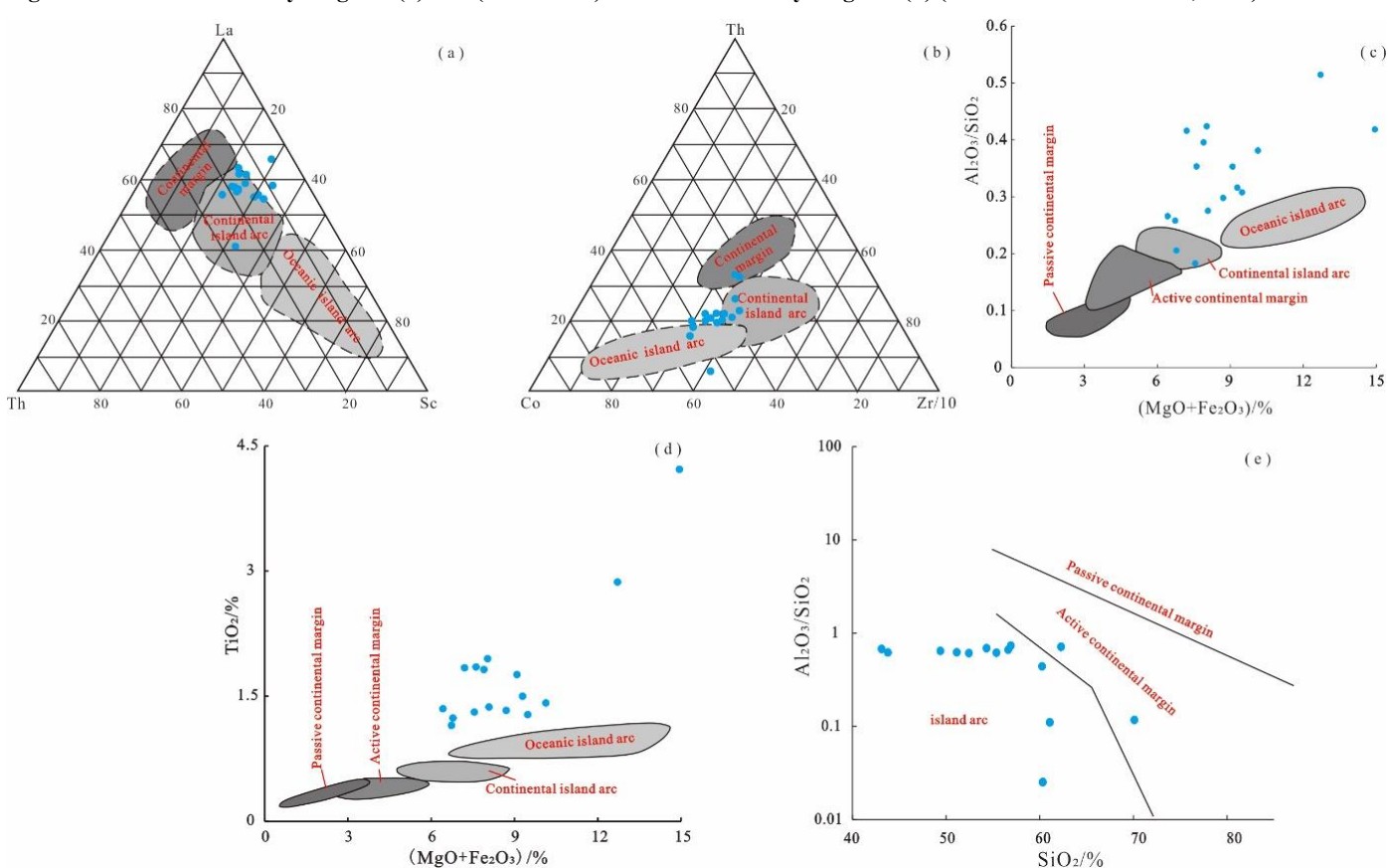


**Figure 11: Tectonic setting of source area in Taodonggou Group mudstone: (a) La-Th-Sc ternary diagram (modified after Zhu et al., 2021);**

**(b) Th-Co-Zr/10 ternary diagram (modified after Zhu et al., 2021); (c) cross plot of Al₂O₃/SiO₂ and Fe₂O₃+MgO (modified after Bhatia,**

**1983); (d) cross plot of TiO₂ and Fe₂O₃+MgO (modified after Bhatia, 1983); (e) cross plot of SiO₂ and Al₂O₃/SiO₂ (modified after Roser**

**and Korsch, 1988)**

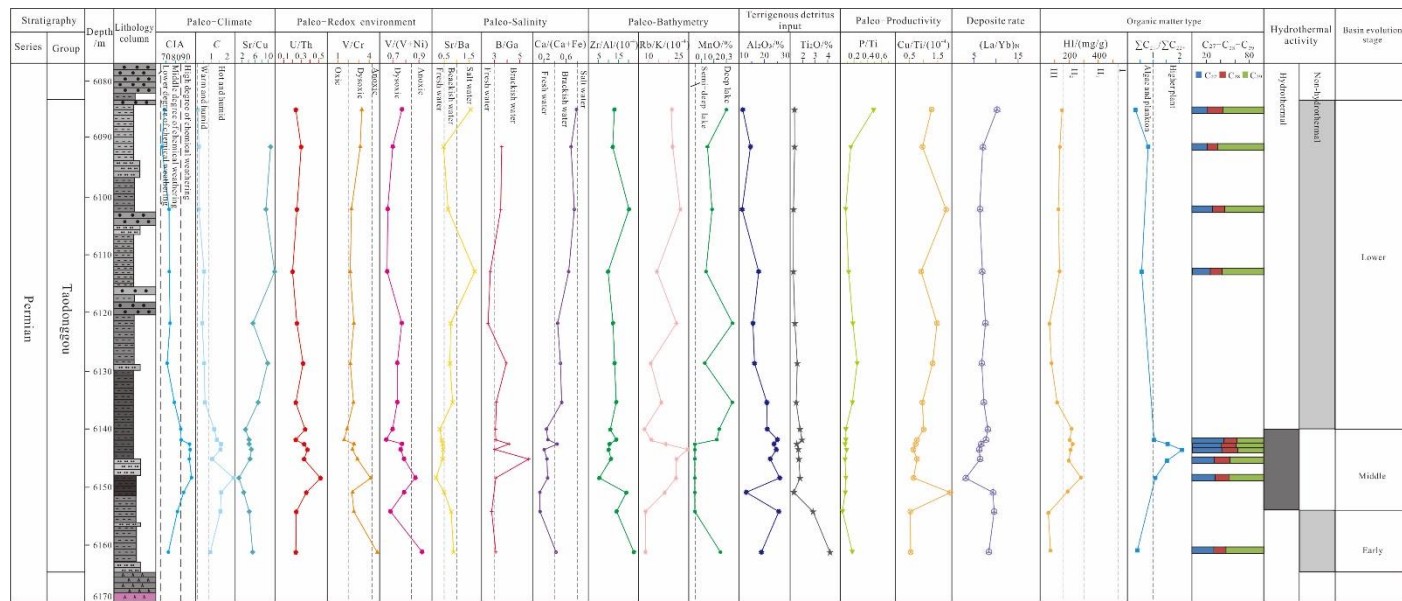


**Figure 12: The geochemical profile of the Taodonggou Group in YT1 well**

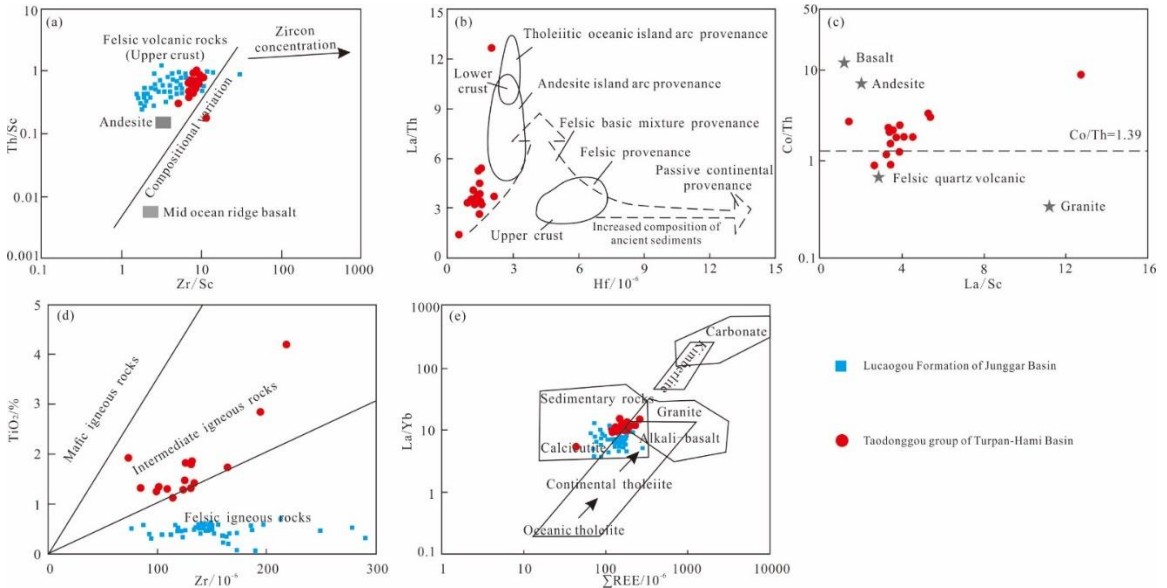


**Figure 13: Parent rock type of Taodonggou Group in YT1 well (Data of Lucaogou Formation in Junggar Basin are from Li et al., 2020):**
**(a) Th/Sc and Zr/Sc intersection diagram(modified after Floyd and Leveridge, 1987); (b) La/Th and Hf intersection diagram(modified**
**after Floyd and Leveridge, 1987); (c) Co/Th and La/Sc intersection diagram(modified after Wronkiewicz and Condie, 1987); (d) TiO₂ and**
**Zr intersection diagram; (e) La/Yb and ∑REE intersection diagram (modified after Allègre and Minster, 1978)**

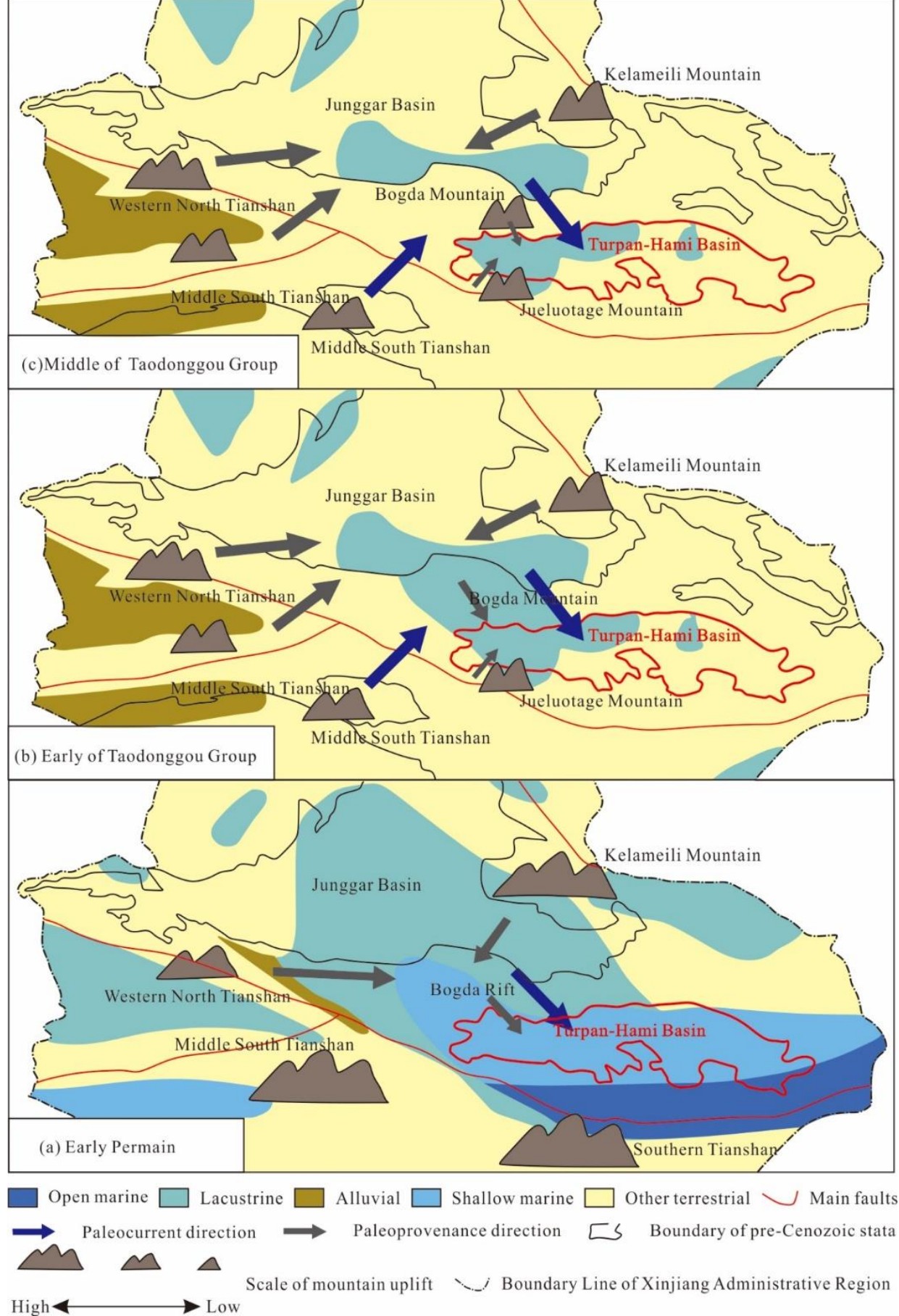

Figure 14: Provenance location from Early Permian to Middle Permian in Tianshan area (modified after Zhao et al., 2020): (a) Early Permian; (b) Early of Taodonggou Group; (c) Middle to later of Taodonggou Group

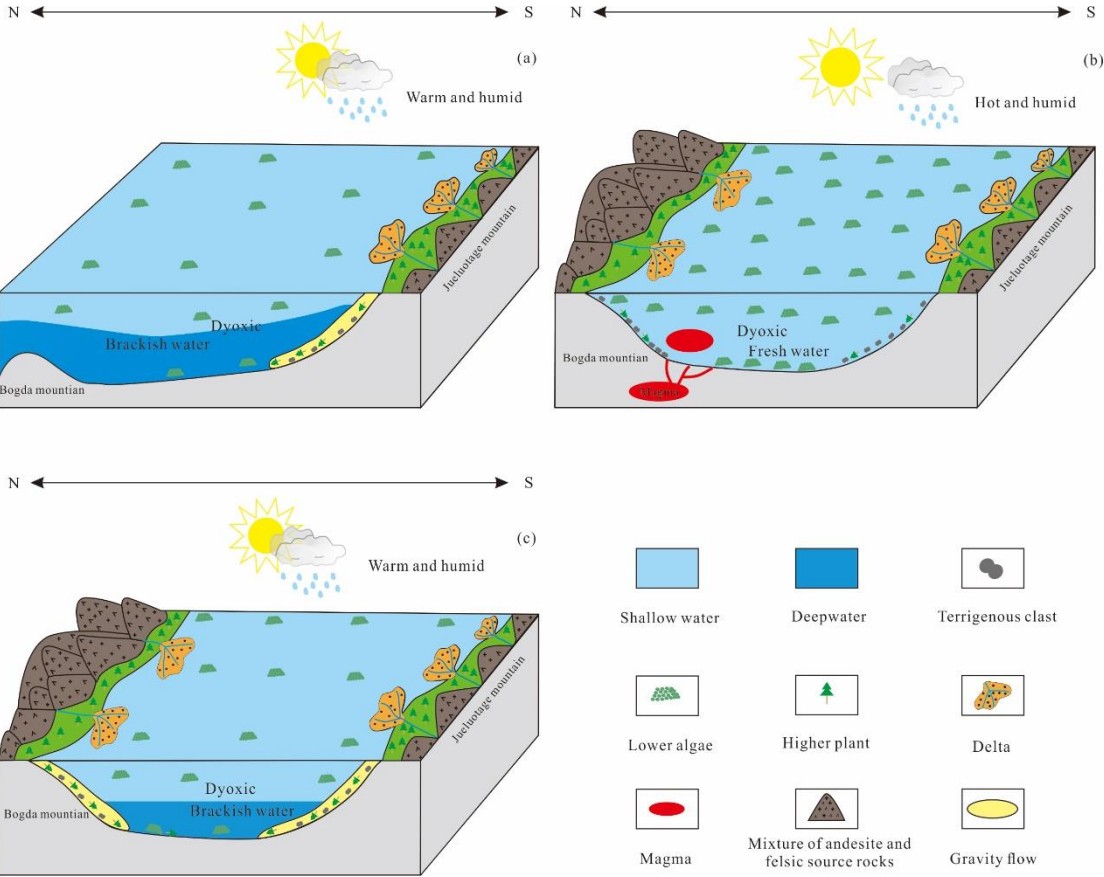


**Figure 15: Middle Permian source sink system and lake basin evolution history of Turpan-Hami basin: (a) Early Taodonggou Group; (b)**
**Middle Taodonggou Group; (c) Late Taodonggou Group**

**Appendix**


Table.A1 Mineral composition of Taodonggou Group mudstone in YT1 well

| Samples | Depth/m | Minerals content/% | | | | | | | |
| --- | --- | --- | --- | --- | --- | --- | --- | --- | --- |
| | | Quartz | K-fledspar | Plagioclace | Calcite | Siderite | Pyrite | Barite | Clay |
| YT1-1 | 6084 | 29.4 | 0.7 | 1.8 | 23.7 | / | / | 13.3 | 31.1 |
| YT1-2 | 6092 | 30.1 | 1.1 | 5.4 | 26.9 | / | / | 2.1 | 34.4 |
| YT1-3 | 6102 | 41.9 | / | 2.5 | 18.1 | / | / | 3.2 | 34.3 |
| YT1-4 | 6113 | 35.7 | 0.5 | 2.9 | 10.8 | / | / | 2.8 | 47.3 |
| YT1-5 | 6122 | 40.5 | 0.1 | 2.7 | 19.4 | / | / | 4.6 | 32.7 |
| YT1-6 | 6129 | 39.2 | 0.6 | 2.1 | 19.7 | / | / | 4 | 34.4 |
| YT1-7 | 6136 | 27.9 | 0.3 | 0.8 | 42.8 | / | / | 4.3 | 23.9 |
| YT1-8 | 6140 | 31.2 | 0.6 | 0.6 | 22.1 | / | / | 5.6 | 39.9 |
| YT1-9 | 6143 | 34 | 0.4 | 0.2 | 14 | / | / | 4.5 | 46.9 |
| YT1-10 | 6144.7 | 33.6 | 2.1 | 3.7 | 11.4 | 2.1 | 2.8 | 3.5 | 40.8 |
| YT1-11 | 6145.3 | 37 | / | 3.7 | 10.6 | 1.4 | / | 2.1 | 45.2 |
| YT1-12 | 6145.8 | 38.4 | 1.7 | 2.2 | 12.5 | / | / | 3.1 | 42.1 |
| YT1-13 | 6147 | 37.9 | / | / | 1 | / | 1.4 | 1.2 | 58.5 |
| YT1-14 | 6151 | 59.2 | 0.5 | 0.2 | 1.3 | / | 2.1 | / | 36.7 |
| YT1-15 | 6154 | 21.8 | 0.8 | 0.8 | 1.8 | / | / | 3.9 | 70.9 |
| YT1-16 | 6161 | 17.2 | 2.3 | 4.8 | 35.4 | 2 | / | 5 | 33.3 |


Table. A2 Major elements of Taodonggou Group mudstone in well YT1

| Samples | Depth/m | Content/% | | | | | | | | | | CIA | P/Ti | $K_2O/Al_2O_3$ |
| --- | --- | --- | --- | --- | --- | --- | --- | --- | --- | --- | --- | --- | --- | --- |
| | | $SiO_2$ | CaO | $Al_2O_3$ | $Fe_2O_3$ | $K_2O$ | $TiO_2$ | $Na_2O$ | MgO | $P_2O_5$ | MnO | | | |
| YT1-1 | 6084 | 43.79 | 19.05 | 11.65 | 5.32 | 3 | 1.35 | 1.15 | 1.1 | 0.9 | 0.3 | 68.71 | 0.49 | 0.26 |
| YT1-2 | 6092 | 54.32 | 14.01 | 14.96 | 6.74 | 3.39 | 1.37 | 1.5 | 1.34 | 0.29 | 0.15 | 70.1 | 0.15 | 0.23 |
| YT1-3 | 6102 | 56.63 | 14.36 | 11.66 | 5.42 | 3.38 | 1.24 | 1.23 | 1.36 | 0.16 | 0.19 | 66.63 | 0.09 | 0.29 |
| YT1-4 | 6113 | 56.92 | 7.38 | 17.52 | 7.93 | 4.2 | 1.28 | 1.22 | 1.55 | 0.21 | 0.14 | 72.55 | 0.12 | 0.24 |
| YT1-5 | 6122 | 51.15 | 12.62 | 15.25 | 7.55 | 3 | 1.33 | 1.2 | 1.15 | 0.3 | 0.34 | 73.85 | 0.17 | 0.20 |
| YT1-6 | 6129 | 62.28 | 4.49 | 16.07 | 5.93 | 3.5 | 1.15 | 1.68 | 0.8 | 1.17 | 0.12 | 70.08 | 0.74 | 0.22 |
| YT1-7 | 6136 | 52.44 | 9.31 | 16.57 | 8.63 | 2.54 | 1.5 | 1.55 | 0.66 | 0.37 | 0.34 | 74.57 | 0.18 | 0.15 |
| YT1-8 | 6140 | 55.37 | 3.01 | 21.11 | 9.64 | 2.63 | 1.42 | 1.5 | 0.49 | 0.15 | 0.24 | 78.92 | 0.08 | 0.12 |
| YT1-9 | 6143 | 60.24 | 2.76 | 21.27 | 8.73 | 1.92 | 1.76 | 0.84 | 0.36 | 0.23 | 0.22 | 85.5 | 0.09 | 0.09 |
| YT1-10 | 6144.7 | 61.08 | 2.75 | 24.16 | 7.54 | 0.99 | 1.82 | 0.3 | 0.36 | 0.21 | 0.06 | 93.83 | 0.08 | 0.04 |
| YT1-11 | 6145.3 | 61.02 | 2.94 | 25.39 | 6.84 | 0.59 | 1.84 | 0.31 | 0.36 | 0.26 | 0.06 | 95.45 | 0.10 | 0.02 |
| YT1-12 | 6145.8 | 60.32 | 5.41 | 21.32 | 7.29 | 0.72 | 1.85 | 0.34 | 0.32 | 0.21 | 0.06 | 93.84 | 0.08 | 0.03 |
| YT1-13 | 6147 | 60.76 | 1.83 | 25.75 | 7.68 | 0.68 | 1.95 | 0.19 | 0.35 | 0.25 | 0.05 | 96.07 | 0.09 | 0.03 |
| YT1-14 | 6151 | 70.11 | 2.44 | 12.83 | 7.28 | 0.97 | 1.31 | 0.34 | 0.27 | 0.15 | 0.05 | 88.59 | 0.09 | 0.08 |
| YT1-15 | 6154 | 49.39 | 1.92 | 25.41 | 12.25 | 2.84 | 2.87 | 1.57 | 0.46 | 0.15 | 0.06 | 80.97 | 0.04 | 0.11 |
| YT1-16 | 6161 | 43.11 | 9.56 | 18.04 | 14.17 | 2.83 | 4.22 | 1.9 | 0.77 | 1.03 | 0.25 | 73.12 | 0.18 | 0.16 |


Table.A3 Characteristics of Trace elements in Taodonggou Group mudstone

| Samples | | YT1-1 | YT1-2 | YT1-3 | YT1-4 | YT1-5 | YT1-6 | YT1-7 | YT1-8 | YT1-9 | YT1-10 | YT1-11 | YT1-12 | YT1-13 | YT1-14 | YT1-15 | YT1-16 |
|---|---|---|---|---|---|---|---|---|---|---|---|---|---|---|---|---|---|
| Depth/m | | 6084 | 6092 | 6102 | 6113 | 6122 | 6129 | 6136 | 6140 | 6143 | 6144.7 | 6145.3 | 6145.8 | 6147 | 6151 | 6154 | 6161 |
| | Be | 0.952 | 1.12 | 1.67 | 1 | 1.52 | 1.26 | 1.74 | 2.17 | 1.79 | 1.31 | 1.35 | 1.42 | 0.711 | 1.77 | 2.05 | 1.55 |
| | Sc | 9.02 | 11.9 | 15.5 | 11.7 | 13.6 | 13.1 | 15.6 | 16.2 | 21.2 | 11.4 | 13.2 | 12.3 | 7 | 24 | 26 | 17.6 |
| | V | 87.3 | 64.2 | 72 | 59.5 | 106 | 89.2 | 100 | 88.5 | 88.7 | 122.3 | 114.6 | 131.6 | 177 | 145 | 124 | 199 |
| | Cr | 27.4 | 21.2 | 31.8 | 27.3 | 40.1 | 43 | 40.1 | 45.1 | 54.8 | 48.5 | 47.6 | 44.5 | 43 | 63 | 51 | 40.2 |
| | Co | 9.46 | 12.4 | 14.9 | 13.3 | 11.9 | 12.6 | 13.7 | 18.2 | 27.6 | 22.3 | 21.7 | 20.6 | 18.7 | 24.8 | 36.9 | 30.4 |
| | Ni | 25.5 | 27.8 | 36.7 | 32.5 | 32.6 | 33.2 | 37.8 | 37.2 | 47.5 | 36.8 | 35.7 | 34.6 | 27.3 | 41.7 | 55.4 | 17.9 |
| | Cu | 34.2 | 33.1 | 44.8 | 34.6 | 51.2 | 41.8 | 39.9 | 50.8 | 48.9 | 52.6 | 50.3 | 51.4 | 57.3 | 55.8 | 64.4 | 71.2 |
| | Zn | 125 | 79 | 92.4 | 96.1 | 69.8 | 90.4 | 74.4 | 67.9 | 78.7 | 64.6 | 63.2 | 65.8 | 44.1 | 70.4 | 114 | 218 |
| Content/ppm | Ga | 17.81 | 20.1 | 23.3 | 19 | 24.8 | 21.9 | 15.1 | 13.2 | 16.1 | 14.5 | 12.7 | 13.7 | 7.14 | 12.7 | 19.2 | 17.3 |
| | Rb | 54.5 | 64.6 | 73.5 | 50.4 | 60 | 33.5 | 33.9 | 16 | 17.9 | 15.6 | 14.6 | 14.6 | 13.6 | 14.2 | 21 | 20.7 |
| | Sr | 758 | 357 | 420 | 422 | 291 | 414 | 269 | 151 | 199 | 214 | 244 | 224 | 63.9 | 126 | 263 | 393 |
| | Mo | 1.29 | 0.661 | 0.866 | 1.24 | 1.09 | 1.44 | 1.17 | 1.23 | 2.68 | 3.02 | 2.88 | 3.14 | 3.86 | 2.14 | 1.18 | 1.28 |
| | Ba | 483.83 | 735.7 | 633 | 547.28 | 159.24 | 547.66 | 326.49 | 465.2 | 565 | 503 | 516 | 505 | 427 | 254 | 303.56 | 424.35 |
| | B | 46.31 | 71.3 | 81.4 | 67.4 | 64.5 | 55.4 | 60.2 | 41.3 | 49.6 | 44.6 | 52.6 | 41.4 | 41.5 | 39.7 | 56.2 | 54.1 |
| | Th | 9.16 | 6.03 | 7.31 | 6.43 | 8.38 | 12.4 | 10.3 | 10.4 | 10 | 9.12 | 8.33 | 8.86 | 6.17 | 7.32 | 9.96 | 3.13 |
| | U | 2.13 | 1.8 | 1.83 | 2.14 | 1.66 | 3.1 | 3.17 | 2.43 | 2.06 | 3.1 | 3.06 | 2.89 | 3.2 | 2.66 | 2.42 | 0.73 |
| | Zr | 82.7 | 99.3 | 124 | 97.1 | 107 | 112 | 123 | 132 | 162 | 128.8 | 130.2 | 123.6 | 70.8 | 130.4 | 192 | 215 |
| | Hf | 2.6 | 3.77 | 4.29 | 3.51 | 3.87 | 4.03 | 4.45 | 4.76 | 5.52 | 4.76 | 3.94 | 4.01 | 2.23 | 3.21 | 6.13 | 6.29 |
| Sr/Ba | | 1.57 | 0.49 | 0.66 | 0.77 | 1.83 | 0.76 | 0.82 | 0.32 | 0.35 | 0.43 | 0.47 | 0.44 | 0.15 | 0.5 | 0.87 | 0.93 |
| Ga/Rb | | 0.33 | 0.31 | 0.32 | 0.38 | 0.41 | 0.65 | 0.45 | 0.83 | 0.9 | 0.93 | 0.87 | 0.94 | 0.53 | 0.89 | 0.91 | 0.84 |
| B/Ga | | 2.6 | 3.55 | 3.49 | 3.55 | 2.6 | 2.53 | 3.99 | 3.13 | 3.08 | 3.08 | 4.14 | 3.02 | 5.81 | 3.13 | 2.93 | 3.13 |
| Rb/K/($10^{-4}$) | | 21.87 | 22.94 | 26.18 | 14.45 | 24.08 | 11.52 | 16.07 | 7.32 | 11.22 | 18.97 | 29.79 | 24.41 | 24.08 | 17.63 | 8.9 | 8.81 |
| V/Cr | | 3.19 | 3.03 | 2.26 | 2.18 | 2.64 | 2.07 | 2.49 | 1.96 | 1.62 | 2.52 | 2.41 | 2.96 | 4.12 | 2.3 | 2.43 | 4.95 |
| V/(V+Ni) | | 0.77 | 0.7 | 0.66 | 0.65 | 0.76 | 0.73 | 0.73 | 0.7 | 0.65 | 0.77 | 0.76 | 0.79 | 0.87 | 0.78 | 0.69 | 0.92 |
| $C$ | | 0.22 | 0.32 | 0.26 | 0.54 | 0.43 | 0.53 | 0.61 | 1.19 | 1.38 | 1.61 | 1.56 | 1.03 | 2.42 | 1.64 | 1.56 | 0.93 |

Table A4 Enrichment Factors of the Taodonggou Group mudstone after AS transformation

| Samples | $X_{EF}$ | | | | | | | | | | | | | | | | | |
|---|---|---|---|---|---|---|---|---|---|---|---|---|---|---|---|---|---|---|
| | Be | Sc | V | Cr | Co | Ni | Cu | Zn | Ga | Rb | Sr | Mo | Ba | B | Th | U | Zr | Hf |
| YT1-1 | 0.43 | 0.11 | 0.92 | 0.42 | 0.68 | 0.51 | 1.04 | 1.80 | 1.28 | 0.53 | 3.66 | 0.63 | 1.14 | 0.68 | 1.05 | 0.78 | 0.71 | 1.28 |
| YT1-2 | 0.39 | 0.12 | 0.53 | 0.25 | 0.70 | 0.44 | 0.78 | 0.89 | 1.13 | 0.49 | 1.34 | 0.25 | 1.35 | 0.82 | 0.54 | 0.52 | 0.66 | 1.44 |
| YT1-3 | 0.75 | 0.19 | 0.76 | 0.48 | 1.07 | 0.74 | 1.36 | 1.33 | 1.68 | 0.72 | 2.03 | 0.43 | 1.49 | 1.20 | 0.83 | 0.67 | 1.06 | 2.11 |
| YT1-4 | 0.30 | 0.10 | 0.42 | 0.28 | 0.64 | 0.43 | 0.70 | 0.92 | 0.91 | 0.33 | 1.36 | 0.41 | 0.86 | 0.66 | 0.49 | 0.52 | 0.55 | 1.15 |
| YT1-5 | 0.52 | 0.13 | 0.85 | 0.47 | 0.66 | 0.50 | 1.19 | 0.77 | 1.37 | 0.45 | 1.07 | 0.41 | 0.29 | 0.73 | 0.73 | 0.47 | 0.70 | 1.45 |
| YT1-6 | 0.41 | 0.12 | 0.68 | 0.47 | 0.66 | 0.48 | 0.92 | 0.94 | 1.14 | 0.24 | 1.45 | 0.51 | 0.94 | 0.59 | 1.03 | 0.83 | 0.69 | 1.44 |
| YT1-7 | 0.55 | 0.14 | 0.74 | 0.43 | 0.69 | 0.53 | 0.85 | 0.75 | 0.77 | 0.23 | 0.91 | 0.40 | 0.54 | 0.62 | 0.83 | 0.82 | 0.74 | 1.54 |
| YT1-8 | 0.54 | 0.11 | 0.51 | 0.38 | 0.72 | 0.41 | 0.85 | 0.54 | 0.52 | 0.09 | 0.40 | 0.33 | 0.61 | 0.34 | 0.66 | 0.49 | 0.62 | 1.29 |
| YT1-9 | 0.44 | 0.14 | 0.51 | 0.46 | 1.09 | 0.52 | 0.81 | 0.62 | 0.64 | 0.10 | 0.53 | 0.72 | 0.73 | 0.40 | 0.63 | 0.42 | 0.76 | 1.49 |
| YT1-10 | 0.28 | 0.07 | 0.62 | 0.36 | 0.77 | 0.36 | 0.77 | 0.45 | 0.50 | 0.07 | 0.50 | 0.72 | 0.57 | 0.32 | 0.50 | 0.55 | 0.53 | 1.13 |
| YT1-11 | 0.28 | 0.08 | 0.55 | 0.33 | 0.72 | 0.33 | 0.70 | 0.42 | 0.42 | 0.07 | 0.54 | 0.65 | 0.56 | 0.36 | 0.44 | 0.52 | 0.51 | 0.89 |
| YT1-12 | 0.35 | 0.08 | 0.76 | 0.37 | 0.81 | 0.38 | 0.85 | 0.52 | 0.54 | 0.08 | 0.59 | 0.84 | 0.65 | 0.33 | 0.55 | 0.58 | 0.58 | 1.08 |
| YT1-13 | 0.14 | 0.04 | 0.84 | 0.30 | 0.61 | 0.25 | 0.79 | 0.29 | 0.23 | 0.06 | 0.14 | 0.86 | 0.46 | 0.28 | 0.32 | 0.53 | 0.27 | 0.50 |
| YT1-14 | 0.72 | 0.27 | 1.39 | 0.87 | 1.62 | 0.76 | 1.54 | 0.92 | 0.83 | 0.13 | 0.55 | 0.95 | 0.54 | 0.53 | 0.76 | 0.89 | 1.01 | 1.43 |
| YT1-15 | 0.42 | 0.15 | 0.60 | 0.36 | 1.22 | 0.51 | 0.90 | 0.75 | 0.63 | 0.09 | 0.58 | 0.27 | 0.33 | 0.38 | 0.52 | 0.41 | 0.75 | 1.38 |
| YT1-16 | 0.45 | 0.14 | 1.35 | 0.39 | 1.41 | 0.23 | 1.40 | 2.03 | 0.81 | 0.13 | 1.23 | 0.41 | 0.65 | 0.51 | 0.23 | 0.17 | 1.19 | 2.00 |
| Average | 0.41 | 0.12 | 0.73 | 0.40 | 0.87 | 0.44 | 0.93 | 0.79 | 0.75 | 0.20 | 0.91 | 0.56 | 0.68 | 0.50 | 0.59 | 0.55 | 0.68 | 1.29 |

Table A5 Characteristics of REE in Taodonggou Group mudstone

| Samples | Depth/m | Content/ppm | | | | | | | | | | | | | | | | | | (La/Yb)$_N$ |
|---------|---------|------|-------|------|------|------|------|------|------|------|------|------|------|------|-------|---------|---------|--------|--------|------|
| | | La | Ce | Pr | Nd | Sm | Eu | Gd | Tb | Dy | Ho | Er | Tm | Yb | Lu | ∑REE | LREE | MREE | HREE | |
| YT1-1 | 6084 | 31.70 | 57.70 | 6.73 | 28.80 | 5.14 | 1.46 | 5.19 | 0.72 | 3.89 | 0.68 | 2.17 | 0.30 | 2.12 | 0.352 | 146.953 | 124.930 | 17.077 | 4.946 | 10.081 |
| YT1-2 | 6092 | 27.30 | 47.80 | 5.51 | 22.90 | 4.79 | 0.84 | 4.13 | 0.73 | 4.25 | 0.71 | 2.40 | 0.37 | 2.56 | 0.408 | 124.695 | 103.510 | 15.447 | 5.738 | 7.190 |
| YT1-3 | 6102 | 27.30 | 48.30 | 5.62 | 23.10 | 4.71 | 1.32 | 4.17 | 0.80 | 4.63 | 0.88 | 2.68 | 0.41 | 2.89 | 0.464 | 127.271 | 104.320 | 16.511 | 6.440 | 6.369 |
| YT1-4 | 6113 | 26.40 | 45.60 | 5.20 | 22.20 | 4.37 | 0.96 | 4.09 | 0.68 | 3.88 | 0.72 | 2.35 | 0.36 | 2.56 | 0.408 | 119.783 | 99.400 | 14.705 | 5.678 | 6.953 |
| YT1-5 | 6122. | 32.60 | 62.80 | 7.61 | 31.70 | 6.56 | 1.96 | 5.77 | 0.98 | 5.35 | 0.97 | 2.89 | 0.43 | 2.92 | 0.429 | 162.971 | 134.710 | 21.590 | 6.671 | 7.527 |
| YT1-6 | 6129 | 33.10 | 80.10 | 7.48 | 31.70 | 6.19 | 0.62 | 5.98 | 0.99 | 5.58 | 0.99 | 3.01 | 0.50 | 3.31 | 0.564 | 180.108 | 152.380 | 20.345 | 7.383 | 6.742 |
| YT1-7 | 6136 | 33.50 | 66.40 | 7.70 | 31.20 | 6.19 | 1.18 | 5.46 | 0.91 | 5.24 | 0.96 | 3.05 | 0.49 | 3.18 | 0.454 | 165.914 | 138.800 | 19.936 | 7.178 | 7.102 |
| YT1-8 | 6140 | 35.90 | 65.80 | 7.23 | 29.20 | 5.47 | 1.65 | 4.96 | 0.96 | 5.35 | 0.96 | 2.97 | 0.47 | 3.01 | 0.426 | 164.346 | 138.130 | 19.344 | 6.872 | 8.041 |
| YT1-9 | 6143 | 39.00 | 73.40 | 9.60 | 40.00 | 7.18 | 1.44 | 5.64 | 1.02 | 5.91 | 1.08 | 3.45 | 0.52 | 3.41 | 0.519 | 192.169 | 162.000 | 22.270 | 7.899 | 7.711 |
| YT1-10 | 6144.7 | 32.60 | 66.43 | 7.34 | 26.40 | 6.31 | 0.98 | 4.82 | 0.84 | 4.97 | 0.86 | 3.12 | 0.33 | 3.21 | 0.436 | 158.646 | 132.770 | 18.130 | 7.096 | 6.847 |
| YT1-11 | 6145.3 | 27.90 | 62.23 | 5.23 | 23.20 | 5.42 | 1.04 | 4.46 | 0.92 | 5.41 | 0.88 | 2.88 | 0.44 | 3.02 | 0.423 | 143.453 | 118.560 | 17.880 | 6.763 | 6.228 |
| YT1-12 | 6145.8 | 30.20 | 65.60 | 5.64 | 25.40 | 5.93 | 1.02 | 5.01 | 0.47 | 4.54 | 0.91 | 2.94 | 0.46 | 3.01 | 0.501 | 151.631 | 126.840 | 5.531 | 6.911 | 6.764 |
| YT1-13 | 6147 | 8.84 | 16.80 | 1.75 | 6.90 | 1.39 | 0.30 | 1.32 | 0.27 | 1.87 | 0.39 | 1.27 | 0.22 | 1.67 | 0.265 | 43.247 | 34.290 | 5.531 | 3.426 | 3.569 |
| YT1-14 | 6151 | 39.40 | 73.60 | 8.64 | 33.60 | 4.22 | 1.84 | 4.32 | 1.21 | 5.83 | 1.03 | 3.42 | 0.43 | 2.98 | 0.392 | 180.912 | 155.240 | 5.531 | 7.222 | 8.914 |
| YT1-15 | 6154 | 52.60 | 105.00 | 12.30 | 50.80 | 9.09 | 2.45 | 7.65 | 1.25 | 7.14 | 1.16 | 3.86 | 0.57 | 3.62 | 0.510 | 257.997 | 220.700 | 28.740 | 8.557 | 9.796 |
| YT1-16 | 6161 | 39.70 | 85.70 | 11.10 | 52.10 | 9.76 | 2.29 | 8.33 | 1.34 | 7.47 | 1.25 | 3.75 | 0.52 | 3.39 | 0.502 | 227.206 | 188.600 | 30.440 | 8.166 | 7.895 |

771     LREE = La + Ce + Pr + Nd; MREE = Sm + Eu + Gd + Tb + Dy + Ho; HREE = Er + Tm + Yb + Lu; (La/Yb)$_N$ = (La/Yb)/(La/Yb)$_{chondrite}$