# Peer review of "Mineralogical and elemental geochemical characteristics of Taodonggou Group"

_EGUsphere, 2022_

## Author Comment (AC3)

Thank you for your thoughtful and hypothetical feedback and reminders. Without your guidance, I may have continued to overlook the serious error of submitting my work to Reviewer 1 in Chinese instead of English. I offer you my sincere apologies and heartfelt appreciation for bringing this to my attention!

The second issue you raised concerns a major drawback in the manuscript, as suggested by Reviewer 2, that I should add evidence of organic geochemistry. At the time, I did not include this section in the paper because the results had already been published in another journal in 2021 and the author's thesis. Therefore, I only included citations in the main text. However, in order to support the existence of Type III kerogen in the deepwater area, I have followed your advice and added HI, as well as the ratio of $\sum C_{21-}/\sum C_{22+}$ and $C_{27}$-$C_{28}$-$C_{29}$. These all serve as evidence that Type III kerogen shale was deposited in a deepwater environment (Fig.1).

Currently, there is almost no dispute that the Taodonggou Group mudstones in the Turpan-Hami Basin were deposited in a deep or semi-deep lake environment, as evidenced by their good response to elemental ratios (Li, 2016; Li, 2019; Song et al., 2018; Xu, 2022). However, your suggestion on the relationship between water depth and organic matter type has greatly enlightened me and broadened my perspective. After conducting a literature review on the existing research in the study area, I did not find any studies on gravity flow. However, by reading studies on gravity flow sedimentation in the Fengcheng Formation shale in the Junggar Basin, there is a possibility of gravity flow sedimentation in the Taibei Sag of the Turpan-Hami Basin. Based on studies by Chen et al. (2003), Xu Haoyu (2022), and Yang Wan et al. (2010; 2017) recommended by Reviewer 1, it is believed that there is sedimentation of coarse clastic rocks at the bottom of the Daheyan Formation (the bottom of the Taodonggou Group), and different sedimentary facies can be identified in different areas of the study area. In addition, during the sedimentation of the Taerlang Formation, three sedimentary facies can be identified around the Bogeda Mountains, including the front edge of the fan delta, turbidite fan, and semi-deep lake-deep lake facies (Wang, 2017), indicating the possibility of gravity flow sedimentation in the study area. The YT1 well is located in the southern part of the Taibei Sag, which has always been the sedimentary center of the Turpan-Hami Basin (Jiang et al., 2015; Li et al., 2021). The mudstones in the study area were deposited in a deep-semi-deep lake environment, with the deeper water depth depositing III-type kerogen mudstones and some layers of relatively thin clastic rocks, suggesting that they were influenced by gravity flows

during the early and late stages. However, I cannot provide actual core evidence for this conclusion because the YT1 well in the study area was drilled deep, and cores were only taken in some depth intervals (6110-6116 and 6140-6154), while other depth intervals were recorded by cuttings. The lithology of the cored intervals is basically consistent. Nevertheless, according to Wang Yue's (2017) study, there is evidence of mixing in the study area, which may prove this point.

Furthermore, I will carefully complete the revisions to the manuscript by systematically incorporating your comments and those of the other two reviewers. Finally, I would like to express my gratitude for your valuable insights. Your comments not only broadened my perspective but also provided important inspiration for my future work. I am truly grateful and have no words to express my appreciation.

[Figure]

Fig. 1 Geochemical Profile of Well YT1

---

## Author Comment (AC4)

[revised manuscript text omitted]

批注 [缪欢5]: Based on the comments of Reviewer 1, the format of the paper has been revised and the results from the discussion section have been moved to the results 
[revised manuscript text omitted]

---

## Author Response (AR1)

**Response to Reviewer 1**

Thank you for your comments and valuable feedback on my manuscript. Based on your feedback and that of two other reviewers, I have made major revisions to my manuscript. The details of the revisions are listed below. To make it easier for you to re-review, I have highlighted the revised sections in yellow in the manuscript.

**Comments:** After going through the introduction of the article, I have a very hard time understand the purpose of this contribution and the authors ignore several decades of research done in the area related to provenance and paleoclimatic reconstructions. I dont see how utilizing 16 samples from a well can therefore help unravel complex tectonic and paleoclimatic processes that are characteristic of the area during the middle Permian.

Based on your feedback, I have rewritten the abstract of the paper as follows:

[revised manuscript text omitted]

As you suggested, I have revamped the title of the manuscript based on the 16 samples and the new title is " *Mineralogical and elemental geochemical characteristics of Taodonggou Group mudstone in Taibei Sag, Turpan-Hami Basin: Implication for its formation mechanism* ".

**Comments:** Very little is mentioned about the stratigraphy of the area, despite the fact of refined stratigraphy for the Taodongou group by Wan Yang and his colleagues. Authors disregard some of the work that Yang and others have done in the area with regards to paleoclimate, provenance, and environmental conditions in the Turpan-Hami Basin.

Thank you very much for your recommendation. Yang et al.'s research in the Turpan-Hami Basin is very detailed, and there exists too much recognizable knowledge in the forty-odd pages of the paper. Therefore, I have combined the results of that paper and some work done by previous authors to summarize the stratigraphic characteristics of the Taodonggou Group

and placed it inside the geological introduction. In addition some gravity flow deposits and other depositional types are also cited by me as depositional modes in the discussion.

**Comments:** In the discussion, the authors start discussing paleoclimate in the region. They go on about their results (which should be included in the results section and not in the discussion) and they have one paragraph that says that they speculate the mudstone was deposited in a warm, humid and hot climate and that these results are similar to those by the same author using biomarkers. This completely disregard previous work in the region and has very minimal discussion on paleoclimate in general for the entire region. What is the novelty and how do these results compare to what has been speculated for the area before? Yang et al. (2010) found a significant amount and well developed calcisols in alluvial fans of the Taodongou group, which would suggest a different paleoclimatic setting than the one discussed here. There are others that have also looked at paleosols (Tabor and his students) which is also disregarded here.

Based on your comments, I have reorganized the conclusion of the article to place the depositional environments based on elemental analyses in the results section, as they should belong in the results. There are many paleoclimate studies on the Middle Permian of the Turpan-Hami Basin, the earliest one is Miao et al. (2004) who judged the climate as warm, humid and hot based on elemental ratios, but his samples are small and there is no discussion on the applicability of the results. Wei et al. (2016) confirmed this based on the age of the trees, and concluded that the Middle Permian of the Turpan-Hami Basin was in the northern subtropical zone of clear seasonality, and the climate was relatively warm and humid, but there were intermittent relatively hot climates. Song et al. (2018) also confirmed that the climate was warm and humid, but there were hot and humid climates based on the Zhaobishan profile and the Taerlang profile. The author proved it again using YT1 well elements and biomarkers. Obviously these are different from Yang et al. study, which considered the climate as semi-arid and semi-humid climate possibly intermittent hot climate. In addition, calcareous nodules do exist at the top of the Taerlang Formation, and oil-bearing calcareous nodules are a typical feature of the Taerlang Formation. However, a warm and humid climate is more favorable for the formation of calcareous nodules. The reason is that (1) warm and humid climate conditions are favorable for the dissolution and deposition of calcium salts in the water body. When water bodies containing calcium ions enter rock crevices or caves, due to changes in environmental conditions, such as temperature, pressure or chemical reactions, calcium ions will precipitate and form calcareous nodules. (2) Calcium nodules in water bodies such as lakes, rivers and oceans are also related to climate. Warm and humid climatic conditions favor the growth of organisms and mineral precipitation in water bodies. Some organisms, such as algae and corals, convert dissolved calcium ions in water into calcium nodules by absorbing them.

Overall, I believe that the Taodonggou Group strata were deposited in a warm, humid and hot climate

**Comments:** The second part of the discussion is the parent rock. The authors also discuss their results and how their results suggest that the parent rocks are andesitic and felsic. But what about the 40 years of work done on the Carboniferous of the

study area? The geological complexity is not discussed and they don't compare the results to those published in the past about provenance. There are very complicated lithologies exposed in both the Tian Shan and Bogda Shan that are Carboniferous in age and I don't think I have seen significant felsitic rocks in the area. Also, these rocks have been buried and dramatically changed tectonically. Can these ratios be influenced by burial processes and postdepositional modification?

There has been a lot of research on the origin of the Turpan-Hami Basin. Currently, it is believed that the Permian origin of the Turpan-Hami Basin mainly comes from the Bogda Mountains in the north and the Jueluotage Mountains in the south, but the main difference lies in the uplift time of the Bogda Mountains. Shao et al. (1999) believed that the origin of the Turpan-Hami Basin is mainly composed of felsic volcanic rocks and andesites. He believed that during the Permian period, the main origin of the Jueluotage Mountains' intermediate acidic igneous rocks was from the uplifted part of the Bogda Mountains, while the uplifted Bogda Mountains were a secondary source, and the main source of the Hami Basin was the Halike Mountains erosion area. In addition, the U-Pb isotope composition measurement results of detrital zircons in the Permian sandstone show good isochron correlation ($R^2$=0.98), and the isochron age is (283 ± 67) Ma, which is consistent with the formation age (268 ± 13) Ma of the source area's granitic rocks within the error range, indicating that the main source of the mineral sand in the Turpan-Hami Basin is from the granitic body in the late stage of the Late Hercynian period, which is the southern source area of the Jueluotage Mountains, thus he believed that the sandstone inherited the origin of the Lower Permian in the Carboniferous. Zhao et al. (2020) analyzed the origin of the Tian Shan region based on a large amount of U-Pb dating data. According to this result, the early origin of the Turpan-hami Basin should be consistent with the Bogda Rift area. However, Song et al. (2018) believed through element analysis that the parent rock types are also andesites and felsic volcanic rocks, mainly from the Bogda Mountains. In the paper by Jonathan Obrist-Farner and Wan Yang (2017), the Permian Quanzijie Formation has neutral rocks, but Jonathan Obrist-Farner and Wan Yang believed that the Upper Quanzijie Formation (260.4-265.8 Ma) is close to the Taodonggou Formation (255-260 Ma) in the manuscript. The preservation degree of the mudstone source of the Taodonggou Group was analyzed using the intersection diagram of Th/Sc and Zr (Figure 13a), and the results showed that the source of the Taodonggou Group mudstone was well preserved and could be used to analyze the provenance information.

**Comments:** The third part of the discussion is related to the uplift of the Bogda Shan. This is still debated and finding that the provenance is different between the Taodongou and Lucaogou Groups/Formations is not sufficient to make the argument about Bogda uplift. Others have argued that these basins were part of a rift system during that time.

Thank you for your comments, However, I hold a different view. Many previous studies have suggested that during the Bogda Rift period, the Junggar Basin, Turpan-hami Basin, Yaggar Basin, and Jimusaer Basin were a single entity with consistent source and organic matter types. This led to early interpretations of the hydrocarbon generation potential and sedimentary environment of the Taodonggou Group using Lucaogou Formation shale from the Junggar Basin or the Jimusaer Basin as examples, as done by Gang Gao et al. (2006) and Shiju Liu et al. (2020). However, their sources are no longer the

same today, which should be able to infer that the Bogda Mountains have already uplifted or partially uplifted. This result is also consistent with the latest research findings (Li et al. (2022) and Wang et al. (2018)).

Li et al. (2022) and Wang et al. (2018) inferred that the Bogda Mountains were uplifted at 289.8 Ma-265.7 Ma, which is significantly earlier than the deposition of the Taodonggou Group (260 Ma-255 Ma), a debate that may be related to Yang et al.'s suggestion that the Taodonggou Group stratigraphy was deposited at 275-294.6 Ma, a stratigraphic time that clearly includes the Turpan-Hami Basin Lower Permian Yierxitu Formation (or Aidinghu Formation). The bottom of the Lower Permian Yierxitu Formation consists of gray-green siltstone and tuffaceous sandstone, and the middle part of the Formation is dominated by yellowish-green and grayish-purple basaltic rocks, locally interspersed with dacite. Tuff and tuffaceous sandstone, and the top is mainly gray-black and black mudstone interbedded with siltstone and sandstone.

**Comments:** Similarly, there is no discussion on the paleosedimentary environment nor the following sections, mainly just description of the results. Dyoxic (should be dysoxic) is also misspelled in the discussion and in the figures. How does this compare to what has already been published in the area? How does it compare to similar sedimentary basins elsewhere?

Thanks again for your comments, the typos I have fixed. The structure of the manuscript has been adjusted, and the article now focuses on the formation mechanism of the Taodonggou Group mudstones in the Turpan-Hami Basin. I have completed the latest version of the manuscript, and I have highlighted the revised parts in yellow.

**Response to Reviewer 2**

Thank you for your comments and valuable feedback on my manuscript. Based on your feedback and the other two reviewers, I have made revisions to my manuscript. In order to make it easier for you to review again, I highlighted the revised parts in blue in the manuscript.

**Comments:** The authors' samples are mudstones, and the results of sandstone research such as shao (1990; 2001) are cited in the discussion section, while the distribution of mudstones in relation to sandstones needs to be added by the authors;

Thanks to your suggestion, I have summarized the relationship between the middle sandstone and mudstone of the Taodonggou Group stratigraphy in a geologic context by summarizing the characteristics of the Taodonggou Group stratigraphy as follows:

*The Daheyan Formation is composed of a sequence of sandstone and conglomerate deposits, with locally interbedded gray to dark gray mudstone. It is unconformably overlain by the Yierxitu Formation. The Taerlang Formation is predominantly composed of gray-black mudstone, with localized occurrences of gray-green siltstone and medium-grained sandstone.*

**Comments:** In the results section, the authors use "~" extensively, and the use of "~" and "-" is different and needs to be adjusted by the authors;

Thank you for your suggestion, I have replaced "~" with "-" in the manuscript as follows:

*The XRD test results of 16 samples from Well YT1 are shown in Table 1 and Figure 2. As can be seen from Table 1 and Figure 2, Taodonggou Group mudstones are composed of clay, quartz, calcite, plagioclase, barite, and K-feldspar, and some samples contain siderite and pyrite. The content of clay is the highest (23.9%–70.9%, mean 40.78%), followed by quartz (17.2%–59.2%, mean 34.69%), calcite (1%–35.4%, mean 16.97%), barite (0%–13.3%, mean 4.21%), plagioclase (0%–5.4, mean 2.93%), and K-feldspar (0%–2.3, mean 0.9%).*

**Comments:** The word is misspelled, "dysoxic" should be rewritten as "dyoxic";

Thanks to your review, I've rewritten "dysoxic" to "dyoxic" throughout the manuscript and highlighted the corrected word in blue.

**Comments:** In provenance, the authors analyzed the provenance of the Turpan-hami Basin Taodonggou Group mudstone and the Junggar Basin Luchaogou Formation mudstone, but they abbreviated the Taodonggou Group mudstone as "P2td" and the Luchaogou Group mudstone as "P2l". This is not recommended, and we suggest the authors change it for international readers;

Thanks to your suggestion, I have removed abbreviations like P2td and P2l from the manuscript, and the relevant text and figures are below:

*At present, there are many opinions about the time of the Bogda Mountain uplift. They think that the initial uplift of Bogda Mountains occurred in Early Permian (Carroll et al., 1990; Shu et al., 2011; Wang et al., 2018; Li et al., 2022), Middle Permian (Zhang et al., 2006; Liu et al., 2018; Wang et al., 2018), Late Permian-Early Triassic (Zhao et al., 2020; Guo et al., 2006; Wang, 1996; Sun and Liu, 2009; Tang et al., 2014; Wang et al., 2018), Middle Triassic (Guo et al., 2006), Early Jurassic (Green et al., 2005; Liu et al., 2017; Ji et al., 2018) and Late Jurassic (Yang et al., 2015). If the initial uplift of the Bogda Mountains was after the middle Permian, the parent rock types of the Taodonggou Group mudstone in the Turpan-Hami Basin and the Luchaogou Formation mudstone in the Junggar Basin should be the same.*

*We have counted the element geochemical characteristics of Luchaogou Formation in the Junggar Basin (Li et al., 2020) and found that the parent rock type of Luchaogou Formation mudstone in the Junggar Basin is greatly different from that of $P_2td$, which is felsic volcanic rock (Fig. 14). As a result, Bogda Mountain's initial uplift should be Late Permian-Early Triassic in the Early Permian or Middle Permian.*

[Figure]

*Figure.13 Parent rock type of Taodonggou Group in YT1 well (Data of Lucaogou Formation in Junggar Basin are from Li et al., 2020): (a)*

*Th/Sc and Zr/Sc intersection diagram(modified after Floyd and Leveridge, 1987); (b) La/Th and Hf intersection diagram(modified after Floyd*

*and Leveridge, 1987); (c) Co/Th and La/Sc intersection diagram(modified after Wronkiewicz and Condie, 1987); (d) TiO$_2$ and Zr intersection*

*diagram; (e) La/Yb and ∑REE intersection diagram (modified after Allègre and Minster, 1978)*

**Comments:** Fig. 8 needs to add a legend, Fig. 11 needs to be adjusted, there are too many blank spaces, and Fig. 14 needs to add organic matter composition and type.

Thanks to your suggestions, I have completed the revision of the figures, which appear in a different order due to the changes made to the structure of the manuscript, as follows:

[Figure]

*Figure.9 Intersection diagram of TiO$_2$ and SiO$_2$ (a) and intersection diagram of Al$_2$O$_3$ and SiO$_2$ (b)*

[Figure]

*Figure.12 The geochemical profile of the Taodonggou Group in YT1 well*

[Figure]

*Figure. 14 Provenance location from Early Permian to Middle Permian in Tianshan area (modified after Zhao et al., 2020): (a) Early Permian;*

*(b) Early of Taodonggou Group; (c) Middle to later of Taodonggou Group*

**Response to Reviewer 3**

Thank you for your comments and valuable feedback on my manuscript. Based on your feedback and that of two other reviewers, I have made major revisions to my manuscript. The details of the revisions are listed below. To make it easier for you to re-review, I have highlighted the revised sections in green in the manuscript.

**Comments:** The other one is that it is my first time to see a paper that proposed source rocks (mudstones) with kerogen type III can be deposited in deep lake facies. Is it really deep lake facies or kerogen type III? I doubt that. I think it is necessary to add evidences from Rock-Eval pyrolysis (vertical variations in HI) and sedimentary analysis (from core observation or

references) to support this viewpoint as well as the lake basin evolution mode in the section 5.5. Because I remain suspicious of this mode. For example, although the authors used a plenty of ratios of elements to support their opinions on paleo-bathymetric variation, it seems that the lithology column of well YT1 shows an opposite variation. Why the dark black mudstones can be deposited in shallower water environment (middle stage), whereas the grey mudstones were deposited in deeper water environment. Is there gravity flows influencing lithology change and source rock quality in the studied area? Again, sedimentary analysis and Rock-Eval pyrolysis are vital and indispensable for basin evolution reconstruction and source rock formation, but they are absent.

Thank you for your comments. I have added evidence of organic geochemistry. This section was not added before because the research results have been published in another journal and the author's paper in 2021. Therefore, I only included citations in the main text. However, in order to support the existence of Type III kerogen in the deepwater area, I have followed your advice and added HI, as well as the ratio of ∑C21-/∑C22+ and C27-C28-C29. These all serve as evidence that Type III kerogen shale was deposited in a deepwater environment (Fig.12).

[Figure]

*Figure.12 The geochemical profile of the Taodonggou Group in YT1 well*

In addition, I have now identified the evidence for gravity flow deposition and have made the necessary changes based on your suggestions. The latest revised section I have nearly placed below. And this section I have highlighted in green text in the revised manuscript.

*5.3 sedimentation mode*

*In previous studies, scholars have believed that the sedimentation of the Permian in the Turpan-Hami Basin is mainly controlled by traction currents (Chen et al., 2003). However, recent research has revealed the presence of gravity flow deposits in the Permian of the Turpan-Hami Basin (Wang et al., 2017; Wang et al., 2018; Xu, 2022). Yang et al. (2010) found poorly sorted debris flow deposits in the Daheyan Formation, and Xu (2022) discovered alluvial and fluvial facies in the Daheyan Formation, consisting of volcaniclastic rocks and conglomerates that are similar in composition to the Lower Permian volcaniclastic rocks and conglomerates. This suggests the existence of gravity flow deposits during the early Permian in the*

*Turpan-Hami Basin. Wang et al. (2018) also suggested the development of gravity flow deposits and pillow lavas in the Early Permian. Meanwhile, in the early Middle Permian, the sedimentation inherited the provenance and sedimentation style from the early Permian, but the gravity flow deposits transitioned gradually into traction current deposits. Due to the influence of gravity flow deposits, terrestrial organic matter can be transported to the deep lake area (Yu et al., 2022; Li et al., 2011), thereby altering the type of organic matter.*

*During the middle of the Taodonggou Group, the Turpan-Hami Basin entered the foreland basin sedimentation stage due to the uplift of the Bogda Mountains. The sedimentary environment of the Taodonggou Group in the Tainan Sag is similar to that in the Taibei Sag (Li, 2019). During this time, the sedimentary water body of the Taodonggou Group in the Turpan-Hami Basin became shallower, and the dominant sedimentation style transitioned to traction currents. Xu (2022) conducted lithological observations on the Taerlanggou section, the Zhaobishan section, and the Y well in the Taodonggou Group and found the presence of traction structures of gravity flow origin in the middle and upper parts of the Taerlang Formation. Additionally, a large number of calcareous and iron nodules appeared in the formation, indicating the occurrence of gravity flow deposits during the late-stage sedimentation of the Taodonggou Group. The organic matter type in the mudstones during this period was influenced by gravity flows.*

---

## Author Response (AR2)

Manuscript number: egusphere-2022-1433

Journal: Solid Earth

Dear editors

We would like to thank you for the opportunity to revise and resubmit our manuscript egusphere-2022-1433, entitled "*Mineralogical and elemental geochemical characteristics of Taodonggou Group mudstone in Taibei Sag, Turpan-Hami Basin: Implication for its formation mechanism* " by Miao et al. We found the topic editors' comments to be helpful in revising the manuscript and have carefully considered and responded to each suggestion, corresponding changes to the resubmitted manuscript are highlighted in yellow.

We also included a response to topic editor in which we addressed comments the topic editor, we hope that these modifications can fulfill the requirements to make the manuscript acceptable for publication. Please let us know if you have any concerns about the manuscript and we would like to address them as soon as possible.

Thank you again for your consideration of the revised manuscript.

Sincerely,

Huan Miao

1627765379@qq.com

**Respond to topic editor**

Thank you for reviewing my manuscript and giving me the opportunity to make revisions. I have now made detailed modifications to this manuscript. For your convenience in reviewing it again, I have highlighted the revised parts in yellow. The detailed comments and modifications are as follows:

**Comment:** Figure 1: depth instead of deoth (in the legend located right of the figure); Figure 1: what is "glutenine"? (is that a typo?)

Thank you for reviewing my manuscript. I have now corrected the typo in Figure 1. Additionally, "glutenine" was a spelling mistake, and it should be "medium-grained sandstone". The revised Figure 1 is as follows:

[Figure]

Figure1: Geological overview of the study area (modified after Miao et al., 2021; Miao et al., 2023): (a) Geological background of Turpan-Hami basin; (b) Thickness contour map of Taodonggou Group mudstone in Taibei sag; (c) YT1 stratum of Taodonggou Group

**Comment:** Lines 167-168 : the authors state that the Taodonggou samples do not show enrichments with respect to the global average shale and they state that only Hf shows a moderate (1.3 enrichment); however, in figure 4 several samples are in the range EF 1-2 and Hf attains values slightly above 2; could the authors please clarify this point?

Thank you for reviewing my manuscript. I have now made revisions to the descriptions in the manuscript, displaying the distribution range and average value of Hf content. Additionally, to facilitate the reading of Figure 4, we have added Table 4 (Enrichment Factors of the Taodonggou Group mudstone after AS transformation). The revised text and the newly added table

are as follows:

*Figure 4 and Table 4 presents the enrichment factors of Taodonggou Group mudstone in the study area. It can be seen from Figure 4 and Table 4 that only Hf (0.5-2.11, mean = 1.29) is enriched in the Taodonggou Group mudstone compared with AS, and other elements are no enriched.*

Table 4 Enrichment Factors of the Taodonggou Group mudstone after AS transformation

| Samples | $X_{EF}$ | | | | | | | | | | | | | | | | | |
|---|---|---|---|---|---|---|---|---|---|---|---|---|---|---|---|---|---|---|
| | Be | Sc | V | Cr | Co | Ni | Cu | Zn | Ga | Rb | Sr | Mo | Ba | B | Th | U | Zr | Hf |
| YT1-1 | 0.43 | 0.11 | 0.92 | 0.42 | 0.68 | 0.51 | 1.04 | 1.80 | 1.28 | 0.53 | 3.66 | 0.63 | 1.14 | 0.68 | 1.05 | 0.78 | 0.71 | 1.28 |
| YT1-2 | 0.39 | 0.12 | 0.53 | 0.25 | 0.70 | 0.44 | 0.78 | 0.89 | 1.13 | 0.49 | 1.34 | 0.25 | 1.35 | 0.82 | 0.54 | 0.52 | 0.66 | 1.44 |
| YT1-3 | 0.75 | 0.19 | 0.76 | 0.48 | 1.07 | 0.74 | 1.36 | 1.33 | 1.68 | 0.72 | 2.03 | 0.43 | 1.49 | 1.20 | 0.83 | 0.67 | 1.06 | 2.11 |
| YT1-4 | 0.30 | 0.10 | 0.42 | 0.28 | 0.64 | 0.43 | 0.70 | 0.92 | 0.91 | 0.33 | 1.36 | 0.41 | 0.86 | 0.66 | 0.49 | 0.52 | 0.55 | 1.15 |
| YT1-5 | 0.52 | 0.13 | 0.85 | 0.47 | 0.66 | 0.50 | 1.19 | 0.77 | 1.37 | 0.45 | 1.07 | 0.41 | 0.29 | 0.73 | 0.73 | 0.47 | 0.70 | 1.45 |
| YT1-6 | 0.41 | 0.12 | 0.68 | 0.47 | 0.66 | 0.48 | 0.92 | 0.94 | 1.14 | 0.24 | 1.45 | 0.51 | 0.94 | 0.59 | 1.03 | 0.83 | 0.69 | 1.44 |
| YT1-7 | 0.55 | 0.14 | 0.74 | 0.43 | 0.69 | 0.53 | 0.85 | 0.75 | 0.77 | 0.23 | 0.91 | 0.40 | 0.54 | 0.62 | 0.83 | 0.82 | 0.74 | 1.54 |
| YT1-8 | 0.54 | 0.11 | 0.51 | 0.38 | 0.72 | 0.41 | 0.85 | 0.54 | 0.52 | 0.09 | 0.40 | 0.33 | 0.61 | 0.34 | 0.66 | 0.49 | 0.62 | 1.29 |
| YT1-9 | 0.44 | 0.14 | 0.51 | 0.46 | 1.09 | 0.52 | 0.81 | 0.62 | 0.64 | 0.10 | 0.53 | 0.72 | 0.73 | 0.40 | 0.63 | 0.42 | 0.76 | 1.49 |
| YT1-10 | 0.28 | 0.07 | 0.62 | 0.36 | 0.77 | 0.36 | 0.77 | 0.45 | 0.50 | 0.07 | 0.50 | 0.72 | 0.57 | 0.32 | 0.50 | 0.55 | 0.53 | 1.13 |
| YT1-11 | 0.28 | 0.08 | 0.55 | 0.33 | 0.72 | 0.33 | 0.70 | 0.42 | 0.42 | 0.07 | 0.54 | 0.65 | 0.56 | 0.36 | 0.44 | 0.52 | 0.51 | 0.89 |
| YT1-12 | 0.35 | 0.08 | 0.76 | 0.37 | 0.81 | 0.38 | 0.85 | 0.52 | 0.54 | 0.08 | 0.59 | 0.84 | 0.65 | 0.33 | 0.55 | 0.58 | 0.58 | 1.08 |
| YT1-13 | 0.14 | 0.04 | 0.84 | 0.30 | 0.61 | 0.25 | 0.79 | 0.29 | 0.23 | 0.06 | 0.14 | 0.86 | 0.46 | 0.28 | 0.32 | 0.53 | 0.27 | 0.50 |
| YT1-14 | 0.72 | 0.27 | 1.39 | 0.87 | 1.62 | 0.76 | 1.54 | 0.92 | 0.83 | 0.13 | 0.55 | 0.95 | 0.54 | 0.53 | 0.76 | 0.89 | 1.01 | 1.43 |
| YT1-15 | 0.42 | 0.15 | 0.60 | 0.36 | 1.22 | 0.51 | 0.90 | 0.75 | 0.63 | 0.09 | 0.58 | 0.27 | 0.33 | 0.38 | 0.52 | 0.41 | 0.75 | 1.38 |
| YT1-16 | 0.45 | 0.14 | 1.35 | 0.39 | 1.41 | 0.23 | 1.40 | 2.03 | 0.81 | 0.13 | 1.23 | 0.41 | 0.65 | 0.51 | 0.23 | 0.17 | 1.19 | 2.00 |
| Average | 0.41 | 0.12 | 0.73 | 0.40 | 0.87 | 0.44 | 0.93 | 0.79 | 0.75 | 0.20 | 0.91 | 0.56 | 0.68 | 0.50 | 0.59 | 0.55 | 0.68 | 1.29 |

**Comment:** Lines 172-177 please put elemental concentrations in this paragraph (and in the whole paper) in ppm units and check the number of digits valid for the normalization ratios; moreover, patterns are not discussed and the significance of the depleted YT1-13 sample is not presented

Thank you for reviewing my manuscript. I have now changed "μg/g" to "ppm" in the manuscript, and I have highlighted these modifications in yellow in the revised version. Additionally, we have added the distribution pattern of REEs and the reason for the depletion and other REEs in sample YT1-13. The REE distribution pattern of sample YT1-13 is consistent with other samples, showing a weakly right dipping REE distribution pattern. However, trace elements in sample YT1-13 are depleted compared to the average shale (highlighted in green in Table 4), indicating that the depletion is caused by groundwater leaching. Therefore, the revised text is as follows:

*The REE content of Taodonggou Group mudstone in the study area is shown in Table 5. According to Table 5, the ∑REE content of Taodonggou Group mudstone ranged from 43.247 ppm to 257.997 ppm, with an average value of 159.206 ppm. The light rare earth element (LREE) content was the highest (mean value 133.45 ppm), followed by medium rare earth element (MREE) (mean value 17.438 ppm) and heavy rare earth element (HREE) (mean value 6.684 ppm) in that order. After chondrite standardization (Taylor and Mclennan, 1985), Taodonggou Group mudstone shows a right dipping REE distribution pattern*

*(Fig. 5), (La/Yb) N is 6.228–10.081, with an average value of 7.358.*

*In addition, in Figure 5, although the YT1-13 sample exhibits a weak right dipping REE distribution pattern similar to other samples, its rare earth elements are significantly depleted. Based on Figure 4 and Table 4, the trace elements in the YT1-13 sample are depleted compared to AS, indicating that the YT1-13 sample has been influenced by groundwater leaching.*

**Comment:** In Figure 6 the chemical alteration index is shown, not the climate index, please modify; moreover, the climate index is presented in the text but not shown, instead the Ga/Rb ratio is presented; how do the C climate index and the Ga/Rb ratio correlate in the studied samples

Thank you for reviewing my manuscript. I have made the necessary revisions to Figure 6, and the climate index C has been added to Table 3. Additionally, I have included the correlations between CIA, C, and Ga/Rb, and updated the corresponding text as follows:

*The CIA values of the Taodonggou Group mudstone in the study area were calculated based on Equation (2) and Equation (3), ranging from 68.71 to 96.97, with a mean value of 80.17. The climate index (C) is 0.22–2.42 (average = 1.01, Tab.3). The overall paleoclimate was warm, humid, and hot (Fig. 6a).*

*In addition, the cross plot of Ga/Rb and $K_2O/Al_2O_3$ can also be used to analyze the paleoclimate characteristics during the formation of sedimentary rocks (Lerman and Baccini, 1987; Liu and Zhou, 2007). As shown in the cross plot of Ga/Rb and $K_2O/Al_2O_3$ (Fig. 6b), almost all points are in the warm/wet area, which indicates that Taodonggou Group mudstone was deposited in a warm and humid paleoclimate.*

*By analyzing the correlations between CIA, C, and Ga/Rb (Figure 6c–e), it can be observed that there is the strongest correlation between CIA and C (Figure 6c, $R^2 = 0.7566$). Additionally, the correlation coefficients between CIA and Ga/Rb, as well as C and Ga/Rb, are both greater than 0.4 (Figures 6d and 6e). This indicates that CIA, C, and Ga/Rb are reliable indicators of the paleoclimate during the sedimentation of the Taodonggou Group mudstone.*

[Figure]

*Figure 6: Paleoclimate of Taodonggou Group: (a) CIA Characteristics of Taodonggou Group mudstone (modified from Nesbitt and Young, 1984); (b) cross plot of $K_2O/Al_2O_3$ and Ga/Rb (modified from Roy and Roser, 2013); (c) cross plot of CIA and C; (d) cross plot of CIA and Ga/Rb; (e) cross plot of C and Ga/Rb*

**Comment:** Fig. 10: "hydrothermal cherts" are written upside down, please modify for better reading

Thank you for reviewing my manuscript. I have made the necessary revisions to Figure 10. The updated Figure 10 is as follows:

[Figure]

*Figure 10: Zn-Ni-Co ternary diagram (a) and (Cu+Co+Ni) ×10-Fe-Mn ternary diagram (b) (modified after You et al., 2019)*